# The increasing atmospheric burden of the greenhouse gas sulfur hexafluoride (SF₆)

Peter G. Simmonds[1], Matthew Rigby[1], Alistair J. Manning[4], Sunyoung Park[8], Kieran M. Stanley[1,10], Archie McCulloch[1], Stephan Henne[2], Francesco Graziosi[11], Michela Maione[11], Jgor Arduini[11], Stefan Reimann[2], Martin K. Vollmer[2], Jens Mühle[3], Simon O'Doherty[1], Dickon Young[1], Paul B. Krummel[5], Paul J. Fraser[5], Ray F. Weiss[3], Peter K. Salameh[3], Christina M. Harth[3], Mi-Kyung Park[9], Hyeri Park[9], Tim Arnold[12,13], Chris Rennick[12], L. Paul Steele[5], Blagoj Mitrevski[5], Ray H. J. Wang[6], and Ronald G. Prinn[7].

[1] School of Chemistry, University of Bristol, Bristol, UK.

[2] Swiss Federal Laboratories for Materials Science and Technology, Laboratory for Air Pollution and Environmental Technology (Empa), Dübendorf, Switzerland.

[3] Scripps Institution of Oceanography (SIO), University of California, San Diego, La Jolla, California, USA.

[4] Met Office Hadley Centre, Exeter, UK.

[5] Climate Science Centre, Commonwealth Scientific and Industrial Research Organisation (CSIRO), Oceans and Atmosphere, Aspendale, Victoria, Australia.

[6] School of Earth, and Atmospheric Sciences, Georgia Institute of Technology, Atlanta, Georgia, USA.

[7] Center for Global Change Science, Massachusetts Institute of Technology, Cambridge, Massachusetts, USA.

[8] Department of Oceanography, Kyungpook National University, Daegu, Republic of Korea.

[9] Kyungpook Institute of Oceanography, Kyungpook National University, Daegu, Republic of Korea.

[10] Institute for Atmospheric and Environmental Sciences, Goethe University Frankfurt, Germany.

[11] Department of Pure and Applied Sciences (DISPEA) of the University of Urbino and Institute of Atmospheric Sciences and Climate (ISAC) of the National Research Council (CNR), Bologna, Italy.

[12] National Physical Laboratory, Teddington, United Kingdom.

[13] School of GeoSciences, University of Edinburgh, Edinburgh, UK.

Correspondence to: P.G. Simmonds (petergsimmonds@aol.com)

**Abstract**

We report a 40-year history of $SF_6$ atmospheric mole fractions measured at the Advanced Global Atmospheric Gases Experiment (AGAGE) monitoring sites, combined with archived air samples to determine emission estimates from 1978-2018. Previously we reported a global emission rate of $7.3 \pm 0.6$ Gigagrams (Gg) yr$^{-1}$ in 2008 and over the past decade emissions have continued to increase by about 24% to $9.04 \pm 0.35$ Gg yr$^{-1}$ in 2018. We show that changing patterns in $SF_6$ consumption from developed (Kyoto Protocol Annex-1) to developing countries (non-Annex-1) and the rapid global expansion of the electric power industry, mainly in Asia, have increased the demand for $SF_6$-insulated switchgear, circuit breakers and transformers. The large bank of $SF_6$ sequestered in this electrical equipment provides a substantial source of emissions from maintenance, replacement and continuous leakage. Other emissive sources of $SF_6$ occur from the magnesium, aluminium, electronics industries and more minor industrial applications. More recently, reported emissions, including those from electrical equipment and metal industries, primarily in the Annex-1 countries, have declined steadily through substitution of alternative blanketing gases and technological improvements in less emissive equipment and more efficient industrial practices. Nevertheless, there are still demands for SF6 in Annex-1 countries due to economic growth, as well as continuing emissions from older equipment and additional emissions from newly installed SF6-insulated electrical equipment, although at low emissions rates. In addition, in the non-Annex-1 countries, $SF_6$ emissions have increased due to an expansion in the growth of the electrical power, metal and electronics industries to support their continuing development.

There is an annual difference of 2.5-5 Gg yr$^{-1}$ (1990-2018) between our modelled top-down emissions and the UNFCCC reported bottom-up emissions, which we attempt to reconcile through analysis of the potential contribution of emissions from the various industrial applications which use $SF_6$. We also investigate regional emissions in East Asia (China, S. Korea) and Western Europe and their respective contributions to the global atmospheric $SF_6$ inventory. On an average annual basis, our estimated emissions from the whole of China are approximately 10 times greater than emissions from Western Europe. In 2018, our modelled Chinese and Western European emissions accounted for ~36% and 3.1 %, respectively, of our global $SF_6$ emissions estimate.

**Keywords**

*Chemical tracers; atmospheric dispersion; models atmospheric transport*

## 1. Introduction

Of all the greenhouse gases regulated under the Kyoto Protocol, $SF_6$ is the most potent, with a global warming potential (GWP) of 23,500 over a 100-year time horizon (Myhre et al., 2013). In practical terms, this high GWP means that 1 ton of $SF_6$ released to the atmosphere is equivalent to the release of 23,500 tonnes of carbon dioxide ($CO_2$). However, the low atmospheric mixing ratio of $SF_6$ relative to $CO_2$ limits its current contribution to total anthropogenic radiative forcing to about 0.2 % (Engel, Rigby, 2019). Nevertheless, with a long atmospheric residence time of 3,200 years, almost all the $SF_6$ released so far will have accumulated in the atmosphere and will continue to do so (Ravishankara et al., 1993).

Vertical profiles of $SF_6$ mixing ratios, collected from balloon flights up to an altitude of about 37 km, indicated that there is very little loss of $SF_6$ due to photochemistry in the troposphere and lower stratosphere (Harnish et al., 1996; Patra et al., (1997). Using an improved atmospheric-chemical-transport model (Patra et al., 2018) reported significantly older 'age of air' (AoA) in the stratosphere and Krol et al., (2018), based on a comparison of six global transport models showed that upper stratospheric AOA varied from 4-7 years among the models. It has been suggested that $SF_6$ may have a shorter atmospheric lifetime ranging from $1937 \pm 432$ years (Patra et al.,1997), 580-1400 years (Ray et al., 2017) and 1120-1475 years (Kovács et al., 2017). However, these shorter, but still very long, $SF_6$ lifetimes would not significantly affect $SF_6$ emissions estimated from atmospheric trends (Engel and Rigby, 2019) .Given the very long lifetime of SF6, compared to the period of our study, uncertainties in this term had a small influence on the outcome. For example, changing the lifetime from 3000 to 1000 years changed the derived emissions by around 1%, which is smaller than the derived uncertainties.

Since the 1970s, $SF_6$ has been used mainly in high voltage electrical equipment as a dielectric and insulator in gas-insulated switchgear, gas circuit breakers, high voltage lines and transformers. Sales compiled from 1996-2003 by producers in Europe, Japan, USA and South Africa (not including China and Russia) showed that, on an annual average basis, 80% of the $SF_6$ produced during this period was consumed by electric utilities and equipment manufacturers for electric power systems (EPA, 2018). Percentage sales, averaged from 1996-2003, for other end-use applications included the magnesium industry (4%), electronics industry (8%), and uses relating to the adiabatic properties of $SF_6$ (3%) e.g., incorporating $SF_6$ into tyres, tennis balls and the soles of trainers as a gas cushioning filler (Palmer, 1996). For example, in 1997 Nike used 277 tonnes (~0.25 Gigagrams, Gg) of $SF_6$ as a filler in its shoes (Harnish and Schwarz, 2003). Other uses in particle accelerators, optical fibre production, lighting, biotechnology, medical refining, pharmaceutical, laboratory, university research and sound-proof windows accounted for around 5% of sales (Smythe, 2004).

Emissions from electrical equipment can occur during production, routine maintenance, refill, leakage, and disposal (Neimeyer and Chu, 1992; Ko et al., 1993). Random failure or deliberate or accidental venting of equipment may also cause unexpected and rapid high levels of emissions. For example, a ruptured seal caused the release of 113 kg of $SF_6$ in a single event in 2013 (Scottish Hydro Electric, 2013). We assume that such random events are generally not recorded when tabulating bottom-up emission estimates, which would lead to an under-estimate in the reported inventories.

Historically, significant emissions of $SF_6$ occurred in magnesium smelting, where it was used as a blanketing gas to prevent oxidation of molten magnesium, in the aluminium industry, also as a blanketing gas, and in semi-conductor manufacturing (Maiss and Brenninkmeijer, 1998). These industries and the electrical power industry accounted for the majority of $SF_6$ usage in the USA (Ottinger et al., 2015). A report on limiting $SF_6$ emissions in the European Union also provided estimates of emissions from sound-proof windows (60 Mg) and car tyres (125 Mg) in 1998, although these applications appear to have been largely discontinued due to environmental concern (Schwarz, 2000).

Sulphur hexafluoride has also been used as a tracer in atmospheric transport and dispersion studies (Collins et al., 1965; Saltzman et al., 1966; Turk et al., 1968; Simmonds et al., 1972; Drivas et al., 1972; Drivas and Shair, 1974). The combined $SF_6$ emissions from reported tracer studies (Martin et al., 2011) were approximately 0.002 Gg. Unfortunately, the amounts of tracer released are often not reported and we conservatively assume that these also amounted to ~ 0.002 Gg, providing a total estimate of about 0.004 Gg (4 tonnes) released from historical $SF_6$ tracer studies. Emissions from natural sources are very small (Busenberg and Plummer, 2000; Vollmer and Weiss, 2002; Deeds et al., 2008).

The earliest measurements of $SF_6$ in the 1970s reported a mole fraction of < 1 pmol mol$^{-1}$ (or ppt, parts per trillion) (Lovelock, 1971; Krey et al., 1977; Singh et al., 1977, 1979). Intermittent campaign-based measurements during the 1970s and 1980s reported an increasing trend. However, it was not until the 1990s that a near-linear increase in the atmospheric burden, throughout the 1980s, was reported (Maiss and Levin, 1994; Maiss et al., 1996, Geller et al., 1997). Fraser et al. (2004), described gas chromatography-electron capture detection (GC-ECD) measurements of $SF_6$ at Cape Grim, Tasmania and noted a long-term trend of 0.1 pmol mol$^{-1}$ yr$^{-1}$ in the late 1970s increasing to 0.24 pmol mol$^{-1}$ yr$^{-1}$ in the mid-1990s. However, after 1995 the annual average growth rate from 1996-2000 declined by 12.5% to 0.21 pmol mol$^{-1}$ yr$^{-1}$, coincident with a ~ 32% decrease in annual sales and prompt releases of $SF_6$ over this same time period (as noted in Table S2 of the Rand report).

Subsequent reports noted a continuing growth in global mole fractions, with an average growth rate of 0.29 ± 0.02 pmol mol$^{-1}$ yr$^{-1}$ after 2000 (Rigby et al., 2010), reaching 6.7 pmol mol$^{-1}$ at the end of 2008 (Levin et al., 2009). This increase in the atmospheric burden of $SF_6$ was also reported by Elkins and Dutton (2009). Measurement of $SF_6$ in the lower stratosphere and upper troposphere was reported to be 3.2 ± 0.5 pmol mol$^{-1}$ at 200 mbar in 1992 (Rinsland et al., 1993). These atmospheric observations have been used to infer global emissions rates ('top-down' estimates). Geller at al., (1997) derived a global emission rate of 5.9 ± 0.2 Gg yr$^{-1}$ in 1996, which by 2008 had increased to 7.2 ± 0.4 Gg yr$^{-1}$ (Levin et al., 2010) or 7.3 ± 0.6 Gg yr$^{-1}$ (Rigby et al., 2010), and to 8.7 ± 0.4 Gg yr$^{-1}$ by 2016 (Engel, Rigby, 2019).

Regional inverse modelling studies indicated that emissions have increased substantially from non-Annex-1 parties to the UNFCCC, particularly in eastern Asia, and that these increases have offset the reduction in emissions from Annex-1 countries (Rigby et al., 2011, 2014; Fang et al., 2014). Rigby et al., (2010) showed an increasing trend in emissions from Asian countries growing from 2.7 ± 0.3 Gg yr$^{-1}$ in 2004–2005 to 4.1 ± 0.3 Gg yr$^{-1}$ in 2008. This rise was large enough to account for all the global emissions growth between these two periods. Similarly, Fang et al. (2014) found that eastern Asian emissions accounted for between 38 ± 5 % and 49 ± 7 % of the global total between 2006 and 2012, with China the

major contributor of emissions from this region. Consistent regional estimates, within the uncertainties, were also reported for China Vollmer et al., (2009); 0.8 (0.53-1.1) Gg yr$^{-1}$ from October 2006-March 2008; Kim et al., (2010); 1.3 (0.23-1.7) Gg yr$^{-1}$ in 2008 and Li et al., (2011); 1.2 (0.9 – 1.7) Gg yr$^{-1}$ from November 2007-December 2008. Emissions from other Asian countries were found to be substantially smaller by Li et al., (2011) with South Korea emitting 0.38 (0.33-0.44) Gg yr$^{-1}$ in 2008 and Japan 0.4 (0.3-0.5) Gg yr$^{-1}$. For North America, $SF_6$ emission estimates of $2.4 \pm 0.5$ Gg yr$^{-1}$ were inferred in 1995 (Bakwin et al., 1997), whereas Hurst et al., (2006) reported emissions of $0.6 \pm 0.2$ Gg yr$^{-1}$ in 2003, consistent with an expectation of declining Annex-1 emissions during this period. Top-down $SF_6$ emissions for Western Europe have been reported by Ganesan et al., (2014), indicating larger modelled emission estimates than those reported to the UNFCCC.

## 1. Methods.

Here, we use a 40-year (1978-2018) time series of $SF_6$ measurements made in situ, and in archived air samples, in combination with a global atmospheric box model and inverse modelling techniques to examine how the growth rate of $SF_6$ has changed, and we estimate global and regional emissions in a top-down approach.

### 1.1 In situ AGAGE measurements

In situ high frequency (every 30 mins) measurements were recorded at Cape Grim, Tasmania beginning in 2001 using a modified Shimadzu gas chromatograph GC fitted with a $Ni^{63}$ electron capture detector ECD, (Fraser et al., 2004). Beginning in 2003, newly developed GC-mass spectrometers (GC-MS) equipped with an automated sample processing system, known as the 'Medusa' were progressively deployed at the AGAGE stations, thereby providing calibrated $SF_6$ measurements every 2 hours (Miller et al., 2008; Arnold et al., 2012). Here we use "Medusa" measurements through 2018, acquired at the five core AGAGE stations; Mace Head, Ireland (beginning in 2003); Trinidad Head, California (beginning in 2005); Ragged Point, Barbados (beginning in 2005); Cape Matatula, American Samoa (beginning in 2006); and Cape Grim, Tasmania (beginning in 2005). At Monte Cimone, Italy (an affiliated AGAGE station), $SF_6$ measurements were measured every 15 minutes using a GC-ECD (Maione et al., 2013). Each real air sample is bracketed with a calibrated (NOAA-2014 scale) air sample analysis resulting in 2 measurements per hour, with a precision of 0.6%.

A complete description of the equipment used in the AGAGE station network is given in Prinn et al., (2000, 2018). We combine these measurements with the Medusa-GC-MS analysis of samples from the Cape Grim Air Archive (GCAA) and a collection of Northern Hemisphere (NH) archived air samples to extend the time series back to 1978 (Rigby et al., 2010). Estimated uncertainties during propagation of calibration standards from Scripps Institution of Oceanography (SIO) to the AGAGE measurement sites was ~0.6% with a calibration scale uncertainty of ~2.0% (Prinn et al., 2018). All archived air and in situ measurements are reported on the SIO-05 calibration scale. The difference between the SIO and National Oceanic and Atmospheric Administration (NOAA) calibration scales is <0.5% (0.03 pmol mol$^{-1}$) (Rigby et al., 2010).

Measurements of $SF_6$ from the UK Deriving Emissions linked to Climate Change network (UK, DECC. https://www.metoffice.gov.uk/research/approach/monitoring/atmospheric-trends/index) started in 2012 at Tacolneston (52.5° N, 1.1° E) and Ridge Hill (52.0° N, 2.5° W), and later in 2013 at Bilsdale (54.4° N, 1.2° W) and Heathfield (51.0° N, 0.2° E), using Agilent GC-ECDs (Stanley et al. 2018; Stavert et al. 2019). At these four sites, $SF_6$ measurements were acquired every 10 minutes and air samples are bracketed with calibrated air samples. In addition to the GC-ECD at Tacolneston, a Medusa GC-MS was installed at the site and has been measuring $SF_6$ since 2012. The GC-ECD at Tacolneston was decommissioned in spring 2018. All calibration gases are on the same scale as the AGAGE stations. Stanley et al., (2018) and Stavert et al., (2019) provide a complete description of the measurement capabilities at the UK sites.

## 2. Bottom-up emission estimates

We compare our model-derived top-down emissions with bottom-up estimates, using reports from the 43 Annex-1 countries that submit annual emissions to the UNFCCC (unfccc.int/process-and-meetings/transparency-and-reporting/reporting-and-review-under-the-convention/greenhouse-gas-inventories-annex-i-parties/national-inventory-submissions-2019, last access: May 1, 2019). This contrasts with the non-Annex-1 countries that are not required to report to the UNFCCC (2010); however, some non-Annex-1 countries do voluntarily submit annual emissions, whereas others report infrequently. For infrequent reporting countries we have linearly interpolated emissions for missing years to provide revised non-Annex-1 emissions. Acknowledging that these bottom-up estimates will have large uncertainties, we see a substantial increase in total emissions from non-Annex-1 countries after 2005, with 50-80% from China. We also compare our estimates with those estimated in EDGAR v4.2 from 1970-2010 (EDGAR, 2010).

In the next section we compile bottom-up emissions estimates based on the usage and release of $SF_6$ in the electrical power, metal and electronics industries. Here we follow the approach of previous publications where $SF_6$ emissions are scaled to electrical production (Fang et al., 2013; Victor and MacDonald, 1999) and attempt to calculate potential emissions from the electrical power industry in China and the rest of the World (ROW) using reported emissions factors for each region.

**3.1 Calculation of $SF_6$ emissions from the electrical power, metal and electronics industries in China and the Rest of the World (ROW).**

3.1.1. Electrical Power

Chinese $SF_6$ emissions, mainly from electrical equipment, account for 60-72% of total emissions from the East Asian region (Fang et al., 2014). Following the method of Zhou et al. (2018), we first determine $SF_6$ consumption (Table 1) from the Chinese electric power industry, using an initial filling factor (FF) of 52 t/GW (range 40-66 t/GW) and then calculate emissions using the highest suggested emission factors, EFs (8.6% manufacture and installation, 4.7% operation and maintenance). For the ROW we also use a median FF of 52t/GW and a 12% loss during manufacture and installation of new equipment and assume

3% loss from banked $SF_6$ in electrical equipment in 1980 and then decreasing linearly to 1%
in 2018, reflecting the change from older to newer equipment, with the reduced leakage of
$SF_6$ (Olivier and Bakker, 1999).

### 3.1.2 Magnesium Industry

256       In the magnesium industry (dye casting, sand casting and recycling), where $SF_6$ is used as
a cover or blanketing gas to prevent oxidation, it is assumed that emissions are equal to
consumption and all the $SF_6$ historically used in the magnesium industry has been emitted
(https://www.ipcc-nggip.iges).  The consumption of $SF_6$ in magnesium production in China
was apparently halted after 2010 and largely replaced with $SO_2$. (National Bureau of
Statistics, 2017). Average annual sales of $SF_6$ to the magnesium industry were estimated to be
~0.25 Gg yr$^{-1}$ from 1996-2003 (Smythe, 2004). Given current regulations and the availability
of substitute blanketing gases and the assumption that China and Russian producers use $SO_2$
as the preferred blanketing gas, we assume that current emissions from the magnesium
industry are equal to or less than the 1996-2003 average of about 0.25 Gg yr$^{-1}$.

### 3.1.3 Aluminium Industry

267       For the aluminium industry, historical emissions of $SF_6$ are poorly understood, as it is
generally assumed to be largely destroyed during the production process by reaction with the
aluminium (Victor and MacDonald, 1994); nevertheless, any surviving $SF_6$ will clearly be
emitted (IPCC, 1997). Maiss and Brenninkmeijer (1998) roughly quantified $SF_6$ consumption
from aluminium degassing (USA and Canada) and $SF_6$-insulated windows (Europe,
predominately Germany) and numerous small specialized applications to about 450 t yr$^{-1}$ in
1995. Since the use of $SF_6$ in these applications have been substantially reduced or eliminated
in the Annex-1 countries, we assume that current global emissions primarily from aluminium
degassing are unlikely to be greater than 0.2 Gg yr$^{-1}$.

### 3.1.4 Electronics Industry

277       Sulphur hexafluoride is used as a general etching agent in the electronics and
semiconductor industry including the production of thin film transistor liquid crystal displays
(TFT-LCDs) and in the cleaning of Chemical Vapor Deposition Chambers (CVD). Fang et.
(2013) reported emissions of 0.15 Gg in 2005 and 0.4 Gg in 2010 from the semiconductor
industry, which has rapidly expanded in China, and emissions from this industry were
reported to be 0.2 – 0.25 Gg yr$^{-1}$ during 2004-2011 (Cheng et al., 2013). Also annual average
consumption by the semiconductor industry from 2012-2018 was reported to be 0.51 Gg yr$^{-1}$
(range 0.41-0.55 Gg), (World semiconductor council, 2020). Due to commercial
confidentiality, there is very little information on the consumption of $SF_6$ in electronics
manufacturing. However, Asian electronics industries, which dominate TFT-LCD
production, have adopted substitute gases, mainly nitrogen trifluoride (NF$_3$), carbon
tetrafluoride (CF$_4$) and HFC-134a (CH$_2$FCF$_3$) in preference to $SF_6$ in recent years. We
therefore assume that global emissions of $SF_6$ from these industries are in the range 0.15 –
0.55 Gg yr$^{-1}$ in the absence of any new information.


### 3.1.5 Production

We also need to consider losses of $SF_6$ that occur during production. Fugitive emissions during $SF_6$ production were estimated to be 0.5% for developed countries (IPCC, 2006). Chinese $SF_6$ production accounts for ~50% of global production and Fang et al. (2013) suggested an EF of 2.2% (1.7-3.3%) for China. We use these EFs for China and an EF of 0.5% for the rest of the world to estimate an average annual $SF_6$ loss from production of ~0.1 Gg yr$^{-1}$ (1990-2018).

The combined emissions from $SF_6$ production, magnesium, aluminium and electronics industries estimated above are approximately 1.1 Gg yr$^{-1}$, which are prone to large uncertainties. This assumes electronics emissions of 0.55 Gg yr$^{-1}$, the highest reported by the World semiconductor council.

## 4. Top-down global emissions estimates

Global emissions were derived using a two-dimensional box model of the atmosphere and a Bayesian inverse method. The AGAGE 12-box model has been used extensively for global emissions estimation and is described in Cunnold et al., (1978, 1996) and Rigby et al., (2013). The model solves for advective and diffusive fluxes between four zonal average 'bands' separated at 30$^o$ north and south and the equator, and between three vertical levels separated at 500 hPa and 200 hPa. A Bayesian inverse modelling approach was adopted that constrained emissions growth rate *a priori*, as described in Rigby et al., (2011; 2014) and used most recently to derive $SF_6$ emissions in Engel, Rigby et al., (2019). Briefly, the approach assumed *a priori* that emissions did not change from one year to the next, with a Gaussian 1-sigma uncertainty in the emissions growth rate set to 20% of the maximum EDGAR v4.2 emissions. The inversion then uses an analytical Bayesian method to find a solution that best fits the observations and this prior constraint. This approach was chosen so that independent constraints on absolute emissions magnitudes (e.g. as in Rigby et al., 2010), which were not available for the entire time period, were not required. Following Rigby et al. (2014), uncertainties applied to the in situ data were assumed to be equal to the variability in the monthly baseline data points, representing the sum of measurement repeatability and a model-data 'mismatch' term parameterising the inability of the model to resolve sub-monthly timescales. For the archive air data, this mismatch uncertainty was taken to have the same relative magnitude as the average mismatch error found during the in situ data period. This term was added to the estimated measurement repeatability of the archive air samples. The influence of these uncertainties, and those of the prior constraint, was propagated through to the *a posteriori* emissions estimate, the uncertainty in which was augmented by an additional term representing the uncertainty in the calibration scale (2%, applied as described in Rigby et al. 2014).

### 4.1. Regional emission estimates using the UK Met Office (InTEM), Empa (EBRIS) and Urbino (FLITS) inverse modelling frameworks.

Three different inverse methods, (1) Inverse Technique for Emission Modelling (InTEM), (2) Swiss Federal Laboratories for Materials Science and Technology (Empa) Bayesian Regional Inversion System (EBRIS), (3) FLexpart Inversion iTalian System (FLITS), Urbino, Italy and two different chemical transport models and were used to estimate regional

SF$_6$ emissions. A brief description of the three inverse methods is given below and a more
detailed description of the InTEM and EBRIS models is provided in the supplementary
information.
InTEM. (Arnold et al., 2018) uses the NAME (Numerical Atmospheric dispersion
Modelling Environment, V7.2) [Jones et al., 2007] atmospheric Lagrangian transport model.
NAME is driven by re-analysis 3-D meteorology from the UK Met Office Unified Model
(Cullen, 1993). We provide estimated emissions for Western Europe (United Kingdom,
Ireland, Benelux countries (Belgium, the Netherlands, and Luxembourg), Germany, France,
Denmark, Switzerland, Austria, Spain, Italy, and Portugal) and, in a separate analysis,
emission estimates for China, using observations recorded at the Gosan station on Jeju Island,
South Korea (33$^o$N, 126$^o$E). Gosan receives air masses mainly from eastern mainland China
during the winter months, with winds from the north-northwest (Rigby et al., 2019; Fang et
al., 2013). We subsequently scale SF$_6$ emissions to a China total by population.
EBRIS. (Henne et al., 2016) employs source sensitivities as derived from the Lagrangian
particle dispersion model FLEXPART (Version 9.1; Stohl et al., 2005) and observed
atmospheric concentrations to optimally estimate spatially resolved surface emissions to the
atmosphere. Here, EBRIS was applied to Western Europe and provided country/region
estimates of *a posteriori* SF$_6$ emissions.
FLITS. (FLexpart Inversion iTalian System. U, Urbino), another modelling approach has
been used for a regional inversion. The model is based on an inversion approach developed by
Stohl et al. (2009). The modelling cascade is composed of the Lagrangian particle dispersion
model (LPDM) FLEXPARTv9.1(http://www.flexpart.eu, downloaded 13 May 2019), in
conjunction with in situ high-frequency observations from four atmospheric monitoring sites
and a Bayesian inversion technique. Here, FLEXPART was driven by operational 3-hourly
meteorological data from the European Centre for Medium-Range Weather Forecasts
(ECMWF) at 1$^\circ \times$1$^\circ$ latitude and longitude resolution, from 2013 to 2018. We run the model in
backward mode, releasing from each measurement sites and every three hours, 40,000 particles
followed backward in time for 20 days. Due to the long atmospheric lifetime of SF$_6$, the model
simulation does not account for atmospheric removal process. For the West European *a priori*
emission field we disaggregated 2 Kt yr$^{-1}$ of SF$_6$ emissions within each country borders
according to a gridded population density data set (CIESIN, Center for International Earth
Science Information Network, www.ciesin.org), and we set 200% of uncertainty of the
emissions for every grid cells. Parametrisation details used here are described in Graziosi et
al., 2015.
In Table 2 we provide details of the East Asian setup of the inversion system (InTEM) and in
Table 3 we provide details of the European setup of the inversion systems (InTEM, EBRIS,
FLITS).

## 5.  Results


Figure 1 and Table 4 shows the AGAGE SF$_6$ mole fractions from 1978-2018, averaged
into semi-hemispheres. In the lower panel of Fig.1, we report the annual SF$_6$ growth rate
increasing from $0.097 \pm 0.013$ pmol mol$^{-1}$ yr$^{-1}$ in 1978 to reach an early maximum average
growth rate in 1995 of $0.24 \pm 0.01$ pmol mol$^{-1}$ yr$^{-1}$ (a Kolmogorov-Zurbenko filter was used
to estimate annual mean growth rates, as described in Rigby et al. (2014). The growth rate
then gradually drops to $0.19 \pm 0.01$ pmol mol$^{-1}$ yr$^{-1}$ in 2000, before increasing to reach $0.36 \pm$
$0.01$ pmol mol$^{-1}$ yr$^{-1}$ in 2018. Between 1978 and 2018, the SF$_6$ loading of the atmosphere has
increased by a factor of about 15. Assuming a radiative efficiency of 0.57 W m$^{-2}$ nmol mol$^{-1}$
(WMO, 2018), SF$_6$ contributed around $5.5 \pm 0.1$ mW m$^{-2}$ in 2018 to global radiative forcing.
In the supplementary material (Fig. S1), we show the model/measurement comparison for the
AGAGE 12-box model.

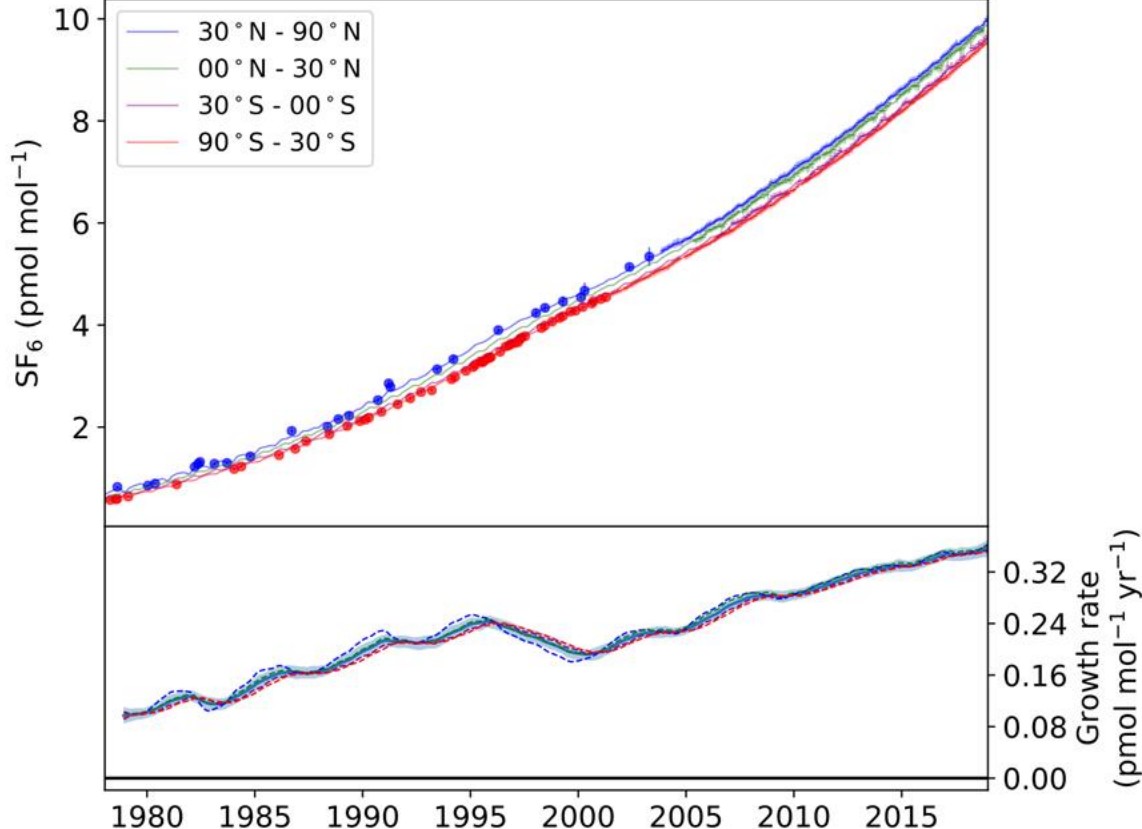


Figure 1. Observed and model-derived SF$_6$ mole fractions and annual growth rates from the
AGAGE 12-box model. Upper panel shows measured atmospheric SF$_6$ mole fractions in each
semi-hemisphere (points with 1-sigma error bars) and archived air samples collected from
1978 in the NH (blue filled circles) and archived air samples collected at Cape Grim
Tasmania in the SH (red filled circles). Solid lines indicate modelled mole fractions using the
mean emissions derived in the global inversion. Semi-hemispheric averages for both the
model and data are shown for 30°–90° N (blue), 0° N–30° N (green), 30° S–0° S (purple) and
90° S–30° S (red). The lower panel shows the model-derived growth rate, smoothed with an
approximately 1-year filter, for each semi-hemisphere (dotted lines), and the global mean and
its 1σ uncertainty (solid line, and shading, respectively).
Our model estimated annual global emissions are shown in Fig. 2 and listed in Table 5.
Here we extend and update the emission estimates prior to 2008, previously described in
Rigby et al. (2010), that reported a global SF$_6$ emission rate of $7.3 \pm 0.6$ Gg yr$^{-1}$ (1-σ
uncertainty unless specified otherwise) in 2008 and Engel, Rigby et al., (2019) estimated
emissions of $8.7 \pm 0.4$ Gg yr$^{-1}$ in 2016. We show that, during the last decade (2008-2018),
emissions have increased by approximately 24%, to $9.04 \pm 0.35$ Gg yr$^{-1}$. At 2018 levels, $SF_6$
emissions are equivalent to $212 \pm 8$ Tg $CO_2$ (assuming a 23,500 100-year global warming
potential, Myhre et al., 2013). Our results demonstrate that, relative to 1978, global $SF_6$
emissions have increased by around 260%, with cumulative global emissions through 2018 of
$234 \pm 7$ Gg ($5500 \pm 170$ Tg $CO_2$-equivalent). Our estimates are in close agreement through
2008 with the independent top-down estimates of Levin et al. (2010). Our estimates show
similar trends to EDGAR v4.2, although our global total is on average 8.9% higher. It should
be noted that the EDGAR estimate includes some information from atmospheric observations
(Rigby et al., 2010). On the other hand, it is likely that Annex-I countries are underreporting
to the UNFCCC (Weiss and Prinn, 2011) and non-Annex-I countries are not required to
report to UNFCCC which explains the much lower UNFCCC totals. There is also close
agreement, within the uncertainties, of our modelled global $SF_6$ emission estimates and those
reported by Krol et al., (2018). The annual average difference was 0.2 Gg yr$^{-1}$ (range 0.01-
0.49 Gg yr$^{-1}$).
Figure 2 and Table 3 record the individual Annex-1 and our revised non-Annex-1
emissions and their combined emissions. UNFCCC emissions reported after 2008 for the
non-Annex-1 countries exceed emissions from Annex-1 countries, as $SF_6$ consumption
moved from Annex-1 countries to non-Annex-1 countries, particularly in Asia. We note that
the significant downward trend in our top-down emission estimate between 1996-2000
matches the UNFCCC reported emissions, furthermore this decline is also consistent with the
drop in sales and prompt emissions listed in the Rand Report (Table S2).
The average annual difference between our global top-down estimates and UNFCCC
reports (Annex-1 plus revised non-Annex-1) listed in Table 5 was 4.5 Gg yr$^{-1}$, reaching a
maximum difference of 5.2 Gg in 2012. This difference subsequently decreased to an annual
average of ~ 4 Gg yr$^{-1}$ between 2013-2018, implying improved or more comprehensive
reporting from non-Annex-1 countries, although we recognise that these differences are prone
to large uncertainties, given the limited emissions data submitted to UNFCCC from the non-
Annex-1 countries.

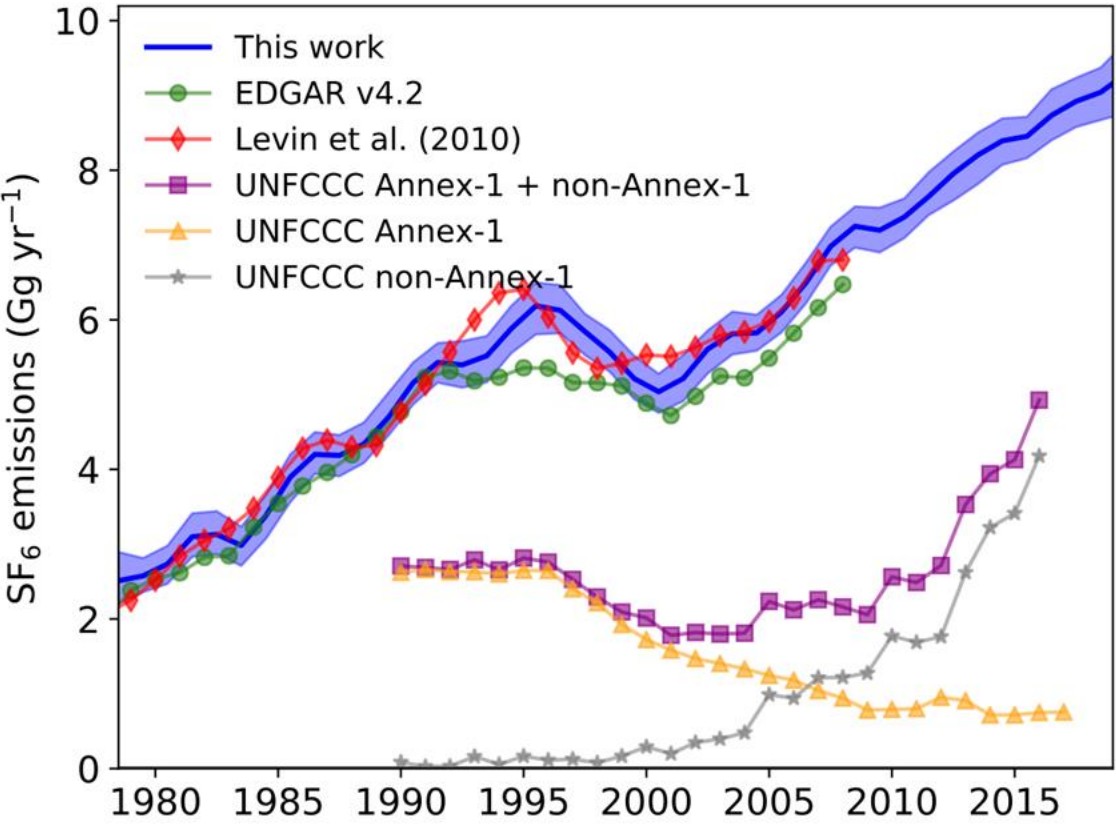

Figure 2. Optimised global SF$_6$ emissions using AGAGE measurements (solid blue line) and shaded line shows the 1σ uncertainties; emissions from Levin et al. (2010, red diamonds); EDGAR v4.2 emissions (green circles); UNFCCC Annex-1 reported emissions (orange triangles); UNFCCC non-Annex-1 reported emissions (grey stars); combined non-Annex-1 and Annex-1 UNFCCC emissions (purple squares).

## 5.1 Regional Emission Estimates

Top-down regional emission estimates have been calculated for two major emission regions of the world, East Asia (China, S. Korea) and North West Europe. As described below, observations from Gosan (Jeju Island, South Korea 33.3° N, 126.2° E) were used to estimate the Chinese emissions, and, for Europe, observations from the UK DECC network and three European AGAGE stations were used (Mace Head, Ireland, MHD; Jungfraujoch, JFJ, Switzerland; and Monte Cimone CMN, Italy). Figure 3 records the high frequency mole fractions of SF$_6$ measured at two AGAGE sites, MHD (53° N, 10° W) and Gosan, GSN, Jeju Island, South Korea (33° N, 126° E). Compared to Mace Head, the Gosan data show very large enhancements (10-30 ppt, compared to 1–2 ppt) above the background mixing ratio of ~5-10 ppt, reflecting significant regional emissions. The Gosan enhancements are associated with the transport of polluted air masses from the north-east part of China, the Korean Peninsula and Japan (Kim et al., 2010; Fang et al., 2014).

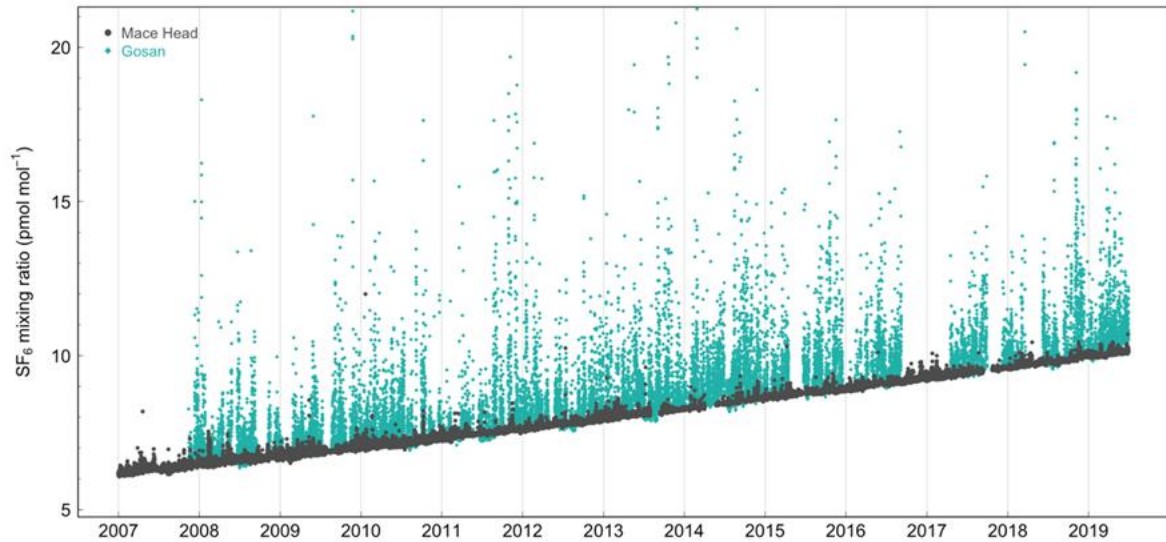

Figure 3. Atmospheric mixing ratios (ppt) recorded at Mace Head, Ireland (black) are shown on top of the measurements at Gosan, Jeju Island, South Korea (green). Elevated mixing ratios represent pollution events associated with regional emissions.

Note: GSN occasionally shows lower mixing ratios than MHD during the summer months when the monsoon transports oceanic background air from the Southern regions to the Gosan site on Jeju Island, South Korea, which accounts for the cases in which GSN mole fractions are lower than those at MHD.

## 5.2 East Asian estimated emissions

Regional top-down estimated emissions for Eastern mainland China, inferred using InTEM and Gosan measurements, are shown in Fig. 4 and listed in Table 5. Derived Chinese emissions (from an area representing 34% of China's population) were subsequently scaled to the whole country by population. China emissions increased from 1.4 (1.0-1.8) Gg yr$^{-1}$ in 2007 to 3.2 (2.6-3.8) Gg yr$^{-1}$ in 2018, an increase of 130 %. Based on the InTEM regional emission estimates, China accounted for 36% (29-42%) of our model estimated global 2018 emissions. The InTEM results show an increasing emission from China, the temporary rise in the mean value in 2011-2012 needs to be understood within the context of the uncertainty estimates, it is plausible, within 1 sigma, that there was no enhanced emissions during this period. Higher (average 38%, 2006-2012) emissions for China (Fang et al., 2014) have been derived using observations from three stations ( Gosan, South Korea; Hateruma, Japan; Cape Ochi-ishi, Japan) rather than one station (Gosan), coarser and different meteorology, a deta;led spatial prior and solved for the whole of China. Also shown in Fig. 4 are our bottom-up estimated emissions calculated from the usage of $SF_6$ in the electrical power industry (Section 3.1), following the methodology published in Zhou et al. (2018) for different filling factors (FF of 40, 52, and 66 t/GW) and high and low emission factors EFs. The assumed high EFs were 8.6% (manufacture and installation) and 4.7% (operation and maintenance) and low EFs of 1.7% (manufacture and installation) and 0.7% (operation and maintenance),

Our bottom-up estimated emissions, using the high EFs, are generally larger than the bottom-
up estimated China emissions determined by Fang et al. (2013), while China estimates based
on the lower EFs suggested by Zhou et al. (2018), are much lower than the other Chinese
emission estimates. Notably from 2007-2012 the bottom-up estimates, after Zhou et al.,
(2018), with a FF of (52 t/GW) and high EF are in close agreement with the top-down
estimates of Fang et al., (2014), They also agree with our results within uncertainties.
However, after about 2014 the increase in these bottom-up estimates especially with the
highest FF (66 t/GW), appear to represent an unrealistically large percentage of global
emissions.
China inventory compiled from biennial submissions to the UNFCCC National
Communications and Biennial Update Report (2018) are also included in Fig. 4 (black filled
circles and interpolated values grey filled circles for missing years). China reported emissions
to the UNFCCC, that were consistently lower than the observation based InTEM modelled
emission estimates through 2012, substantially increased in 2014 to within the uncertainties
of the modelled emissions.

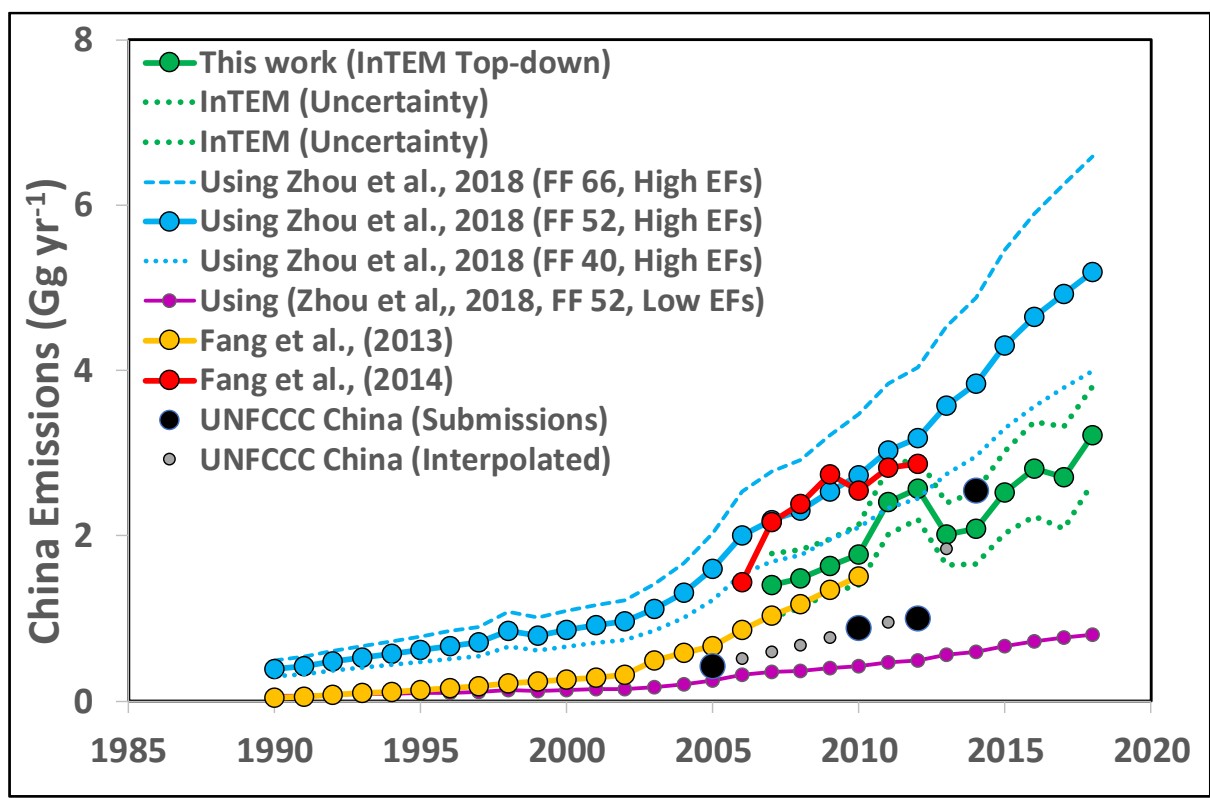


Figure 4. Bottom-up and Top-down emission estimates scaled for the whole of China.

Top-down Chinese emission estimates, Fang et al. (2014), agree within the uncertainties,
with our bottom-up emission estimates. Conversely our top-down InTEM Chinese emission
estimates fall between our bottom-up and the Fang et al. (2013) bottom-up emission
estimates.  Regardless of the EF used, our bottom-up emission estimates would require lower
Chinese EFs to obtain closer agreement with our top-down emission estimate.

Figure 5 shows the footprint of the mapped China emission magnitudes determined from
InTEM, based on measurements recorded at the Gosan station, South Korea. Although our
main focus has been on emissions from China, it is clear from Fig. 5 that there are also
emissions from South Korea. The 2007-2018 average annual $SF_6$ InTEM emission estimate
for South Korea (population ~52 M) is $0.26 \pm 0.05$ Gg yr$^{-1}$ with a slight upward trend (+0.007
Gg yr$^{-2}$). This compares well with the reported average value of 0.36 Gg yr$^{-1}$ over the period
2007-2014 with an upward trend of +0.006 Gg yr$^{-2}$ (South Korea, 2017, second biennial
report). The emissions for South Korea are higher per head of population (~0.005 Gg/M) than
those estimated for China (population ~1400 M, ~0.002 Gg/M in 2018). For Western Europe,
discussed in the next section, the equivalent value is ~0.001 Gg/M.
In Supplementary Table S3 we list the InTEM $SF_6$ emission estimates for South Korea.
The average annual emissions of South Korea (0.26 Gg yr$^{-1}$) are similar to those of Western
Europe (0.22 Gg yr$^{-1}$) and in both cases are approximately 1/10 of Chinese average annual
emissions (2.2 Gg yr$^{-1}$).

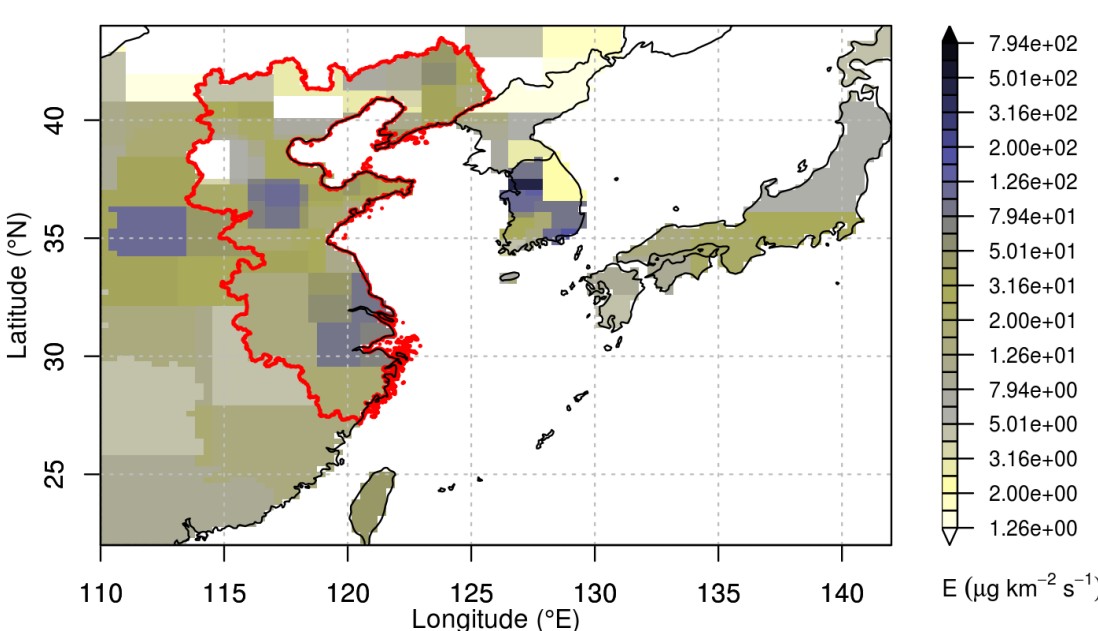

Figure 5. Map of the top-down emission estimate from China and East Asia. The red line
indicates the boundary of the region we denote 'eastern mainland China', to which the
measurements at Gosan and the inversion method are most sensitive.

**5.3 Western Europe emission estimates**

InTEM estimated top-down emissions (2013-2018) for Western Europe (United
Kingdom, Ireland, Benelux, Germany, France, Denmark, Switzerland, Austria, Spain, Italy,
and Portugal) from measurements at 7 sites (Mace Head (MHD), Ireland, Bilsdale, UK
(BSD), Heathfield, UK (HFD), Ridge Hill, UK (RGL) and Tacolneston, UK (TAC),
Jungfraujoch (JFJ), Switzerland, and Monte Cimone (CMN), Italy, are presented in Fig. 6 and
listed in Table 6. EBRIS used observations from 4 sites (MHD, TAC, JFJ, CMN) to estimate
top-down emissions for the period 2013-2018. Emissions from the InTEM and ERBRIS

inversion models are in close agreement with inventory emissions (UNFCCC 2019). FLITS also used observations (2013-2018) from 4 sites (MHD, TAC, JFJ, CMN) and an inverse model to estimate top-down emissions, which are higher than the other two results but follow a similar trend. The emission flux uncertainty decreases from 200% for the *a priori* to ~25 % for the *a posteriori* emission field (average over the study period), supporting the reliability of the results. Top-down emissions for Western Europe from 4 inversion systems for the year 2011 were reported to be 47% higher than UNFCCC, with Germany identified as the principal emitter (Brunner et al., 2017).

The contribution of Western European $SF_6$ emissions to the global total in 2018 was 3.1% (2.4-3.9 %, Table 6, average of all inversions). Comparing the model estimated $SF_6$ emissions from Western Europe and China, it is apparent that China is a much larger contributor to the global $SF_6$ inventory. On an annually averaged basis, top-down Chinese emissions exceed those emitted from Western Europe by a factor of ~10. For Western Europe, EFs are generally expected to be lower, representing better maintenance practices and more efficient $SF_6$ capture during re-filling (EU Commission, 2015). The faster uptake of $SF_6$ substitutes and vacuum-insulated units would also explain the much lower emission estimates in Western European countries.

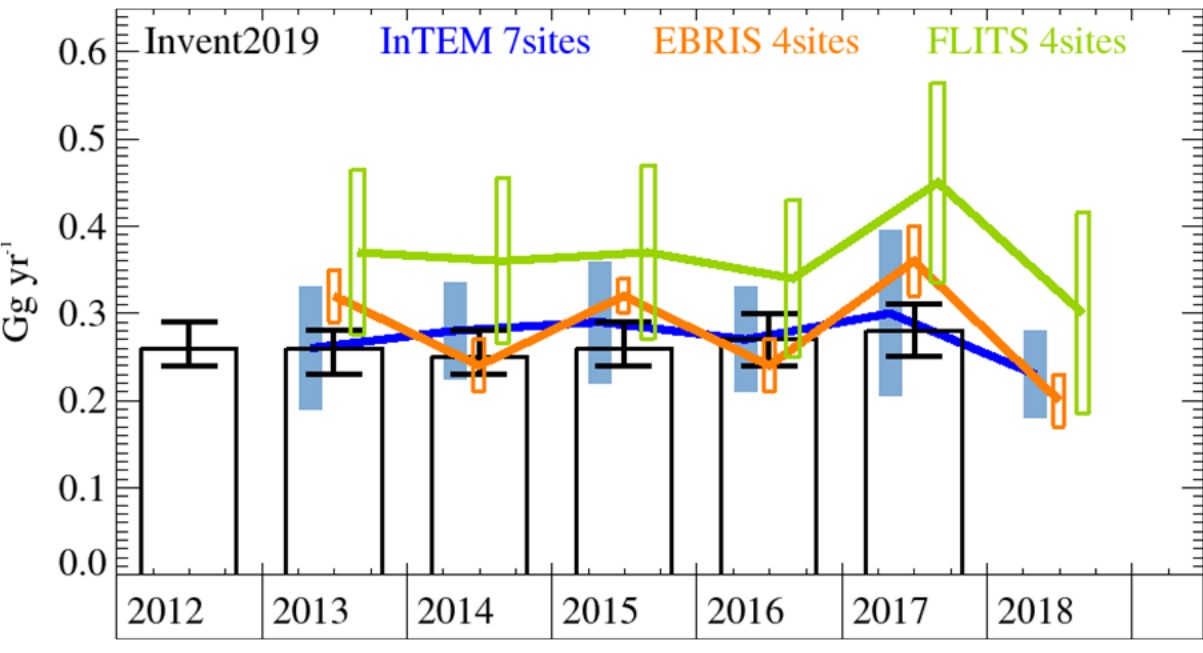

Figure 6. $SF_6$: Western Europe emission estimates (Gg yr$^{-1}$) from the UNFCCC Inventory black; InTEM inversion (2013-2018, blue, 7 sites: MHD, JFJ, CMN, TAC, RGL, HFD, BSD); EBRIS (2013-2018, orange, 4 sites: MHD, TAC, JFJ, CMN). FLITS (2013-2018, green, 4 sites: MHD, TAC, JFJ, CMN).
The uncertainty bars are ± 1 std.

Figure 7 shows the footprint of the average emission estimates for Western Europe calculated using three inverse models (InTEM, EBRIS and FLITS), illustrating that significant emissions are located in southern Germany, a region with a substantial number of semi-conductor producers (https://prtr.eea.europa.eu/#home).

565

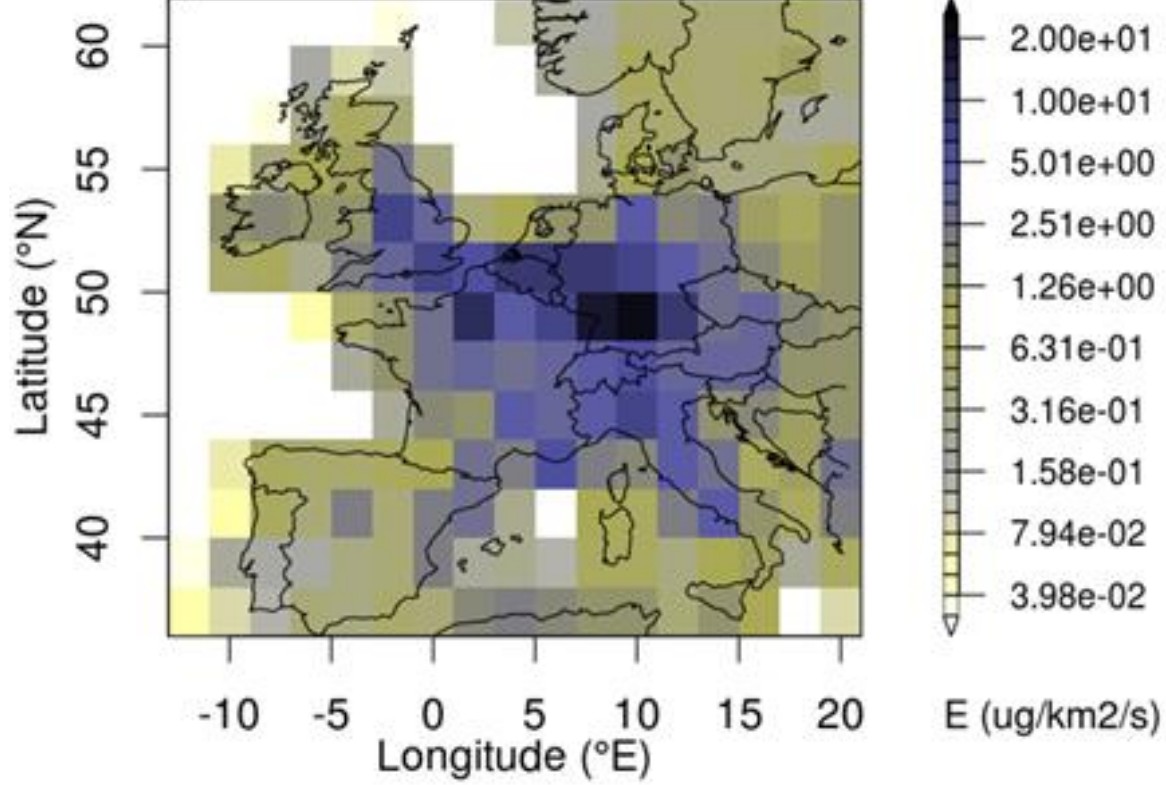

Figure 7. Top-down inversion emission estimate for Western Europe (2013-2018). Average of InTEM (7 observation sites), EBRIS (4 observation sites) and FLITS (4 observation sites).

## 6. Increasing global SF$_6$ emissions and the deficit between bottom-up and top-down emissions estimates

Weiss and Prinn (2011) noted that SF$_6$ bottom-up estimates derived from industrial accounting and reported to the UNFCCC by Annex-1 countries are likely under-reported, actually representing 80% of the total in the mid-1990s and 60% of the total in 2006, leading to poor agreement (under-reported by a factor of 2) with top-down emissions estimates determined from atmospheric observations. However, for Western Europe (Sect.5.3) our estimated emissions (2013-2018) from the model inversions are in close agreement with the UNFCCC reported inventory. Limitations imposed by commercial secrecy and the lack of consistent reporting of SF$_6$ emissions, both from Annex-1 and non-Annex-1countries, continue to contribute to the discrepancies between bottom-up and top-down methods.

We next explore if the increasing global emissions of SF$_6$ may be related to changing patterns of source location and usage in electrical equipment, magnesium smelting, aluminium production and electronics manufacturing, and attempt to reconcile the large average annual discrepancy of ~ 4.5 Gg yr$^{-1}$ between bottom-up and top-down emissions estimates.

Previous reports on SF$_6$ emissions from the electrical power industry have noted that emission factors (EFs) may vary widely depending on the type of equipment and different

maintenance and servicing practices (Capiel/Unipede, 1999). About 12% of $SF_6$ consumed in
the manufacture and commissioning of electrical equipment is estimated to be directly
emitted. Industry assessments of the maximum leakage during operation from older
equipment (manufactured before 1980) was 3% $yr^{-1}$, although higher leakage rates for some
countries continued into the 1990s. For example, in 1995 USA annual refill and leakage of
circuit breakers was estimated to be 20% of total installed stock; however, with the
installation of improved self-contained equipment leakages steadily reduced to 0.5-1% $yr^{-1}$
(Olivier and Bakker, 1999). A recent study of the $SF_6$ losses from gas-insulated electrical
equipment in the UK calculated an average annual leakage rate of 0.46% $yr^{-1}$ from the
inventory of $SF_6$ held in the installed equipment and 1.29% $yr^{-1}$ from the transmission
network, with an overall average (2010-2016) leakage rate of 1% $yr^{-1}$ from the UK electrical
power industry (Widger and Haddad, 2018).
Figure 8 shows the global installed electrical capacity and the percentage contribution of
wind and solar power capacity from 2000-2018. Installed electrical capacity grew by 62%
(2412 GW) during this period. Of this rise, ~45% was due to solar and wind, illustrating the
very rapid growth rate of the renewable sector, as utility companies invested in renewable
energy (GWEC, 2018; CWEA, 2018; IRENA, 2019). The inset panel records the percentage
of solar and wind power by country during 2017 – led by China, USA, and Germany. The
global adoption of renewable technologies, especially hydroelectric, wind and solar power
has been particularly strong in the non-Annex-1 countries to support their continuing
development (Fang et al., 2013). For example, Chinese installed electrical capacity, relative
to the ROW, increased from about 3% in 1980 to ~43% in 2018, as noted in Table 1.
We assume that with the wider geographical distribution of renewables, compared with
the localised gas or oil-fired power stations, this has resulted in many more connections to the
electricity grid and a consequent rise in the number of gas-insulated electrical switches,
circuit breakers, and transformers. With the adoption of more technologically advanced GIS
with lower emissions we might expect there to be a reduction in overall $SF_6$ emissions over
time. There is clearly a balance between the very substantial increase in the global number of
newly installed GIS equipment and the major advances in reducing the leakage of $SF_6$ from
GIS equipment and the recovery and substitution of $SF_6$. At present the larger number of
global GIS installations appear to be overpowering the success in reducing $SF_6$ emissions.

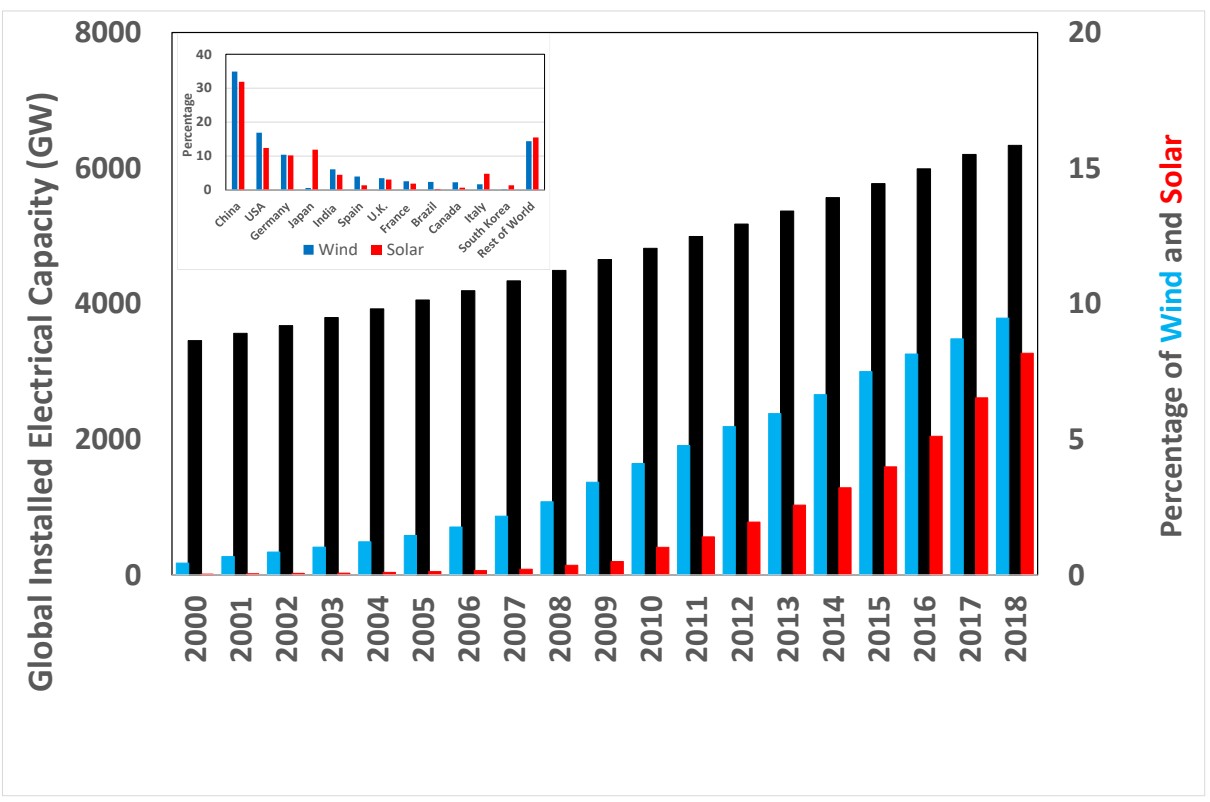

Figure 8. Global installed electrical capacity (GW) and the percentage contribution from wind power (blue bars) and from solar power (red bars) from 2000-2018. Insert: Percentage of wind and solar power by country in 2017 (IRENA, 2019).

Sulphur hexafluoride in the electrical power industry is primarily used in high voltage gas-insulated switchgear (GIS) which consumes > 80% of the $SF_6$ used, with medium voltage GIS consuming only about 10% (Niemeyer and Chu, 1992; Dervos and Vassiliou, 2000; Zu et al., 2011; Xiao et al., 2018). Since this electrical equipment can be operational for 30-40 years, there is a large bank of $SF_6$ in older equipment that will be a continuing source of global $SF_6$ emissions through routine maintenance, decommissioning, catastrophic failure of components (as noted previously) and long-term leakage. Sulphur hexafluoride is also used by the utility companies in gas-insulated transmission lines (Ecofys, 2018).

Recently, regulations have been introduced to mitigate the environmental impact of $SF_6$ emissions. The European Commission reinforced a 2006 F-Gas regulation in 2015 (No. 517/2014) with the aim of reducing the EU's F-gas emissions by two-thirds from 2014 levels by 2030 (EU Commission, 2015). It is important to realise that under these current European regulations there are no restrictions on the use of $SF_6$ in switchgear, but there are requirements to recover $SF_6$ where possible (Biasse, 2014). Historically, $SF_6$ has been the preferred insulator and arc-quenching gas, although technological advances and alternative gases to $SF_6$ have been introduced to reduce overall emissions. Substitute gases include perfluoroketones, perfluoronitriles and trifluoroiodomethane ($CF_3I$) (Okubo, 2011; Li et al., 2018, Xiao et al., 2018). Some wind turbine manufacturers have recently started to offer $SF_6$-free equipment or vacuum insulated switch gear. In 1995 the U.S. Environmental Protection Agency (EPA) established an $SF_6$ Emissions Reduction Partnership for electric power systems to improve equipment reliability and reduce $SF_6$ emissions by technological

innovation. They subsequently reported that by 2016 there had been a 74% reduction of $SF_6$
emissions by the industrial partners (EPA, 2018).
The combined bottom-up emissions estimated in Sect. 3.1 from $SF_6$ production and the
various industrial applications which use $SF_6$ are ~1.1 Gg $yr^{-1}$. Sales of $SF_6$ to these industries
are listed in the Rand report (Smythe, 2004) from 1996-2003 (Supplement: Table S2) which
does not include recent data and only covers an unspecified part of the globe, implying a
potential underestimation of actual emissions. Assuming $SF_6$ consumption from these
industries are emitted promptly (i.e. not banked), we calculate an average annual emission
from 1996-2003 of 1.1 Gg $yr^{-1}$ (0.83-1.42 Gg $yr^{-1}$), that includes the magnesium, electronics,
adiabatic and the fraction of 'other uses' that are emitted promptly. The agreement between
our bottom-up estimate of industrial emissions and the estimate derived from sales are
consistently lower than modelled top-down emission estimates. Even accepting these many
assumptions and uncertainties the dominant emissions are attributable to the electrical
industry and its use of $SF_6$ insulated equipment.
We can also obtain an 'effective' EF for the electrical industry by first subtracting prompt
emissions from our top-down emission estimate and then calculating the amount of $SF_6$
required to match the remaining annual top-down emission estimate, given global installed
electrical capacity and an assumed FF (52t/GW; Zhou et al., 2018). Based on this simplified
method we estimate an 'effective' average EF of 2.5% for the entire time period. In Figure 9
we compare our top-down emissions estimate and a bottom-up estimate with $\pm$ 25%
uncertainty (prompt emissions + electrical industry emissions using a median FF of 52 t/GW
and the inferred effective EF). Notwithstanding some disagreement between the top-down
estimate and the simple bottom-up model during certain periods (e.g. an overestimate in the
early 1980s and underestimate during the 1990s, it is notable that decadal trends in $SF_6$ can be
broadly explained by the rise in installed electrical capacity and a single effective EF. This
suggests that, when considered on ~10 year timescales, reductions in EF achieved in certain
countries through new technologies or improved GIS management, have been offset by the
growth in higher-EF GIS from other parts of the world, such that the effective EF has not
changed substantially on a global scale.

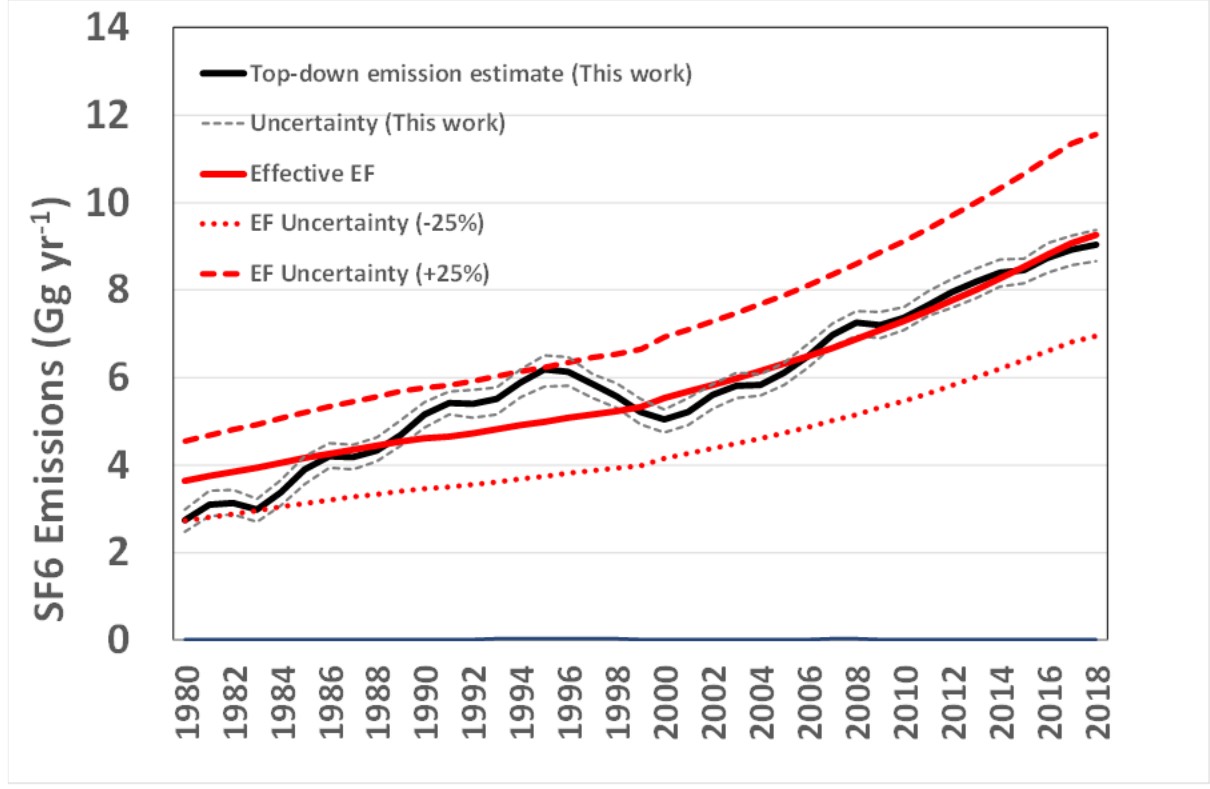

Figure 9. Top-down emission estimate (solid black line) with 1-sigma uncertainties and a bottom-up estimate, based on $SF_6$ prompt emissions plus an emissions estimate from the electrical power industry (FF=52 t/GW) and an inferred effective EF (2.5%) with ±25% uncertainty (dashed lines).

## 4. Conclusions

New atmospheric $SF_6$ mole fractions are presented which extend and update our previously reported time series from the 1970s to 2008 by a further 10 years to 2018 in both hemispheres. We estimate global emissions of $SF_6$ using data from the 5 core AGAGE observing sites and archived air samples with a 12-box global chemical transport model and an inverse method. $SF_6$ emissions exhibited an almost linear increase from 2008-2018 reaching $9.0 \pm 0.4$ Gg yr$^{-1}$ in 2018, a decadal increase of ~24%. Chinese emissions in 2018 based on InTEM regional emission estimates, with large uncertainties, account for 36% (29-42%) of total global emissions relative to our model estimated global 2018 emissions.

We find that on an annually averaged basis emissions from China are about 10 times larger than emissions from South Korea and Western Europe. Relative to 1978, global $SF_6$ emissions have increased by ~260% with cumulative global emissions through December 2018 of $234 \pm 6$ Gg or $CO_2$ equivalents $5.5 \pm 0.2$ Pg. To further mitigate the large uncertainties will require an increase in the number of monitoring sites, improved transport models and a substantial improvement in the accuracy and transparency of emissions reporting.

We note that the rapid expansion of global power demand and the faster adoption of renewable technologies, such as wind and solar capacity over the past decade, particularly in the Asia region, has provided a large bank of $SF_6$ which currently contributes to the

atmospheric burden of $SF_6$ and will continue throughout the lifetime (30-40 years) of the installed equipment. The resultant increase in $SF_6$ emissions from the non-Annex-1 countries has overwhelmed the substantial reductions in overall emissions in the Annex-1 countries, where less emissive industrial practices are used in the handling of $SF_6$ (EPA, 2018; EU Commission, 2015). This also suggests that any decrease in emission factor from Annex-1 countries has been offset by an increase in non-Annex-1 emission factors. The non-Annex-1 countries are progressively using improved and less emissive GIS electrical equipment however, it is only in the last few years that alternative gases to $SF_6$ or $SF_6$-free equipment have been commercially available for switchgear and other electrical systems. We conclude that the observed increase in global installed electrical capacity, in both developed and developing countries, is consistent with the temporal rise in $SF_6$ global emissions.

Table 1. Estimated $SF_6$ emissions (Gg) calculated from China and the Rest of the World (ROW) installed electrical capacity.

| Year | [1]China installed electrical capacity (GW) | [2]China estimated emissions (Gg) | [3]ROW installed electrical capacity (GW) | [4]ROW estimated emissions (Gg) | Global emissions (ROW + China) (Gg) |
|---|---|---|---|---|---|
| 1980 | 66 | 0.17 (0.13-0.21) | 1910.8 | 3.48 (2.79-4.28) | 3.65 (2.93-4.50) |
| 1981 | 69 | 0.18 (0.14-0.23) | 1996.2 | 3.58 (2.75- 4.54) | 3.76 (2.89-4.77) |
| 1982 | 72 | 0.19 (0.15-0.24) | 2068.0 | 3.53 (2.72-4.48) | 3.72 (2.87-4.72) |
| 1983 | 76 | 0.20 (0.16-0.26) | 2133.3 | 3.52 (2.71-4.46) | 3.72 (2.87-4.72) |
| 1984 | 80 | 0.21 (0.16-0.27) | 2225.3 | 3.74 (2.88-4.75) | 3.95 (3.04-5.02) |
| 1985 | 87 | 0.24 (0.19-0.31) | 2303.5 | 3.69 (2.84-4.68) | 3.93 (3.02-4.99) |
| 1986 | 94 | 0.26 (0.20-0.33) | 2368.6 | 3.62 (2.78-4.59) | 3.88 (2.98-4.92) |
| 1987 | 103 | 0.29 (0.23-0.37) | 2430.1 | 3.59 (2.76-4.56) | 3.88 (2.99-4.93) |
| 1988 | 115 | 0.33 (0.26-0.43) | 2486.7 | 3.55 (2.73-4.51) | 3.89 (2.99-4.93) |
| 1989 | 127 | 0.36 (0.28-0.46) | 2555.7 | 3.63 (2.79-4.61) | 3.99 (3.07-5.07) |
| 1990 | 138 | 0.39 (0.30-0.49) | 2594.4 | 3.40 (2.62-4.32) | 3.79 (2.91-4.81) |
| 1991 | 151 | 0.43 (0.33-0.54) | 2617.9 | 3.25 (2.50-4.12) | 3.68 (2.93-4.66) |
| 1992 | 167 | 0.48 (0.37-0.61) | 2661.5 | 3.33 (2.56-4.23) | 3.81 (2.93-4.84) |
| 1993 | 184 | 0.53 (0.40-0.67) | 2712.4 | 3.34 (2.57-4.24) | 3.87 (2.98-4.91) |
| 1994 | 201 | 0.57 (0.44-0.72) | 2766.0 | 3.32 (2.56-4.22) | 3.89 (2.99-4.94) |
| 1995 | 219 | 0.62 (0.47-0.78) | 2800.9 | 3.15 (2.42-4.00) | 3.77 (2.90-4.78) |
| 1996 | 238 | 0.67 (0.51-0.85) | 2858.5 | 3.25 (2.50-4.13) | 3.92 (3.02-4.98) |
| 1997 | 256 | 0.71 (0.54-0.90) | 2905.0 | 3.13 (2.41-3.98) | 3.84 (2.95-4.87) |
| 1998 | 289 | 0.85 (0.66-1.08) | 2922.8 | 2.87 (2.21-3.64) | 3.72 (2.87-4.73) |
| 1999 | 302 | 0.80 (0.61-1.01) | 2984.6 | 3.10 (2.393.94) | 3.90 (3.00-4.95) |
| 2000 | 319.3 | 0.86 (0.66-1.09) | 3135.7 | 3.69 (2.84-4.68) | 4.55 (3.50-5.77) |
| 2001 | 338.5 | 0.91 (0.70-1.16) | 3224.8 | 3.27 (2.52-4.15) | 4.18 (3.22-5.31) |

| | | | | | |
|---|---|---|---|---|---|
| 2002 | 357.6 | 0.96 (0.74-1.22) | 3319.7 | 3.27 (2.52-4.15) | 4.23 (3.26-5.37) |
| 2003 | 392.4 | 1.11 (0.86-1.42) | 3404.8 | 3.16 (2.43-4.02) | 4.27 (3.29-5.43) |
| 2004 | 443.5 | 1.31 (1.01-1.67) | 3479.4 | 3.04 (2.34-3.85) | 4.35 (3.35-5.52) |
| 2005 | 517.8 | 1.60 (1.23-2.03) | 3536.9 | 2.85 (2.19-3.62) | 4.45 (3.42-5.65) |
| 2006 | 624.1 | 2.00 (1.54-2.54) | 3568.8 | 2.59 (1.99-3.29) | 4.59 (3.53-5.83) |
| 2007 | 720.6 | 2.19 (1.69-2.78) | 3617.5 | 2.61 (2.00-3.31) | 4.80 (3.69-6.09) |
| 2008 | 798.5 | 2.30 (1.77-2.92) | 3692.2 | 2.69 (2.07-3.41) | 4.99 (3.84-6.33) |
| 2009 | 883.1 | 2.54 (1.95-3.22) | 3767.2 | 2.61 (2.01-3.31) | 5.15 (3.96-6.53) |
| 2010 | 966.4 | 2.73 (2.10-3.47) | 3850.9 | 2.58 (1.98-3.27) | 5.31 (4.09-6.74) |
| 2011 | 1062.5 | 3.03 (2.33-3.84) | 3929.5 | 2.45 (1.89-3,11) | 5.48 (4.22-6.96) |
| 2012 | 1146.8 | 3.18 (2.45-4.04) | 4027.9 | 2.49 (1.91-3.16) | 5.67 (4.36-7.19) |
| 2013 | 1257.7 | 3.57 (2.75-4.53) | 4107.9 | 2.27 (1.75-2.88) | 5.84 (4.49-7.41) |
| 2014 | 1369.2 | 3.84 (2.96-4.88) | 4195.7 | 2.21 (1.70-2.81) | 6.05 (4.66-7.69) |
| 2015 | 1506.7 | 4.30 (3.31-5.46) | 4265.8 | 1.99 (1.53-2.52) | 6.29 (4.83-7.97) |
| 2016 | 1645.8 | 4.64 (3.57-5.89) | 4343.1 | 1.91 (1.47-2.42) | 6.55 (5.04-8.32) |
| 2017 | 1777.0 | 4.93 (3.79-6.26) | 4424.6 | 1.81 (1.39-2.30) | 6.74 (5.19-8.56) |
| 2018 | 1899.7 | 5.19 (3.99-6.59) | 4435.7 | 1.65 (1.39-1.96) | 6.84 (5.38-8.55) |

721

[1] China installed electrical power capacity compiled from
www.statistica.com/statistics/302269 and www.iea.com (IEA 2017).

[2] estimated China emissions derived from method of Zhou et al. (2018), using an initial
filling of 52 t/GW (in parenthesis, range 40-66 t/GW) and emission factors of 8.6%
(manufacture and installation) and 4.7% (operation and maintenance).

[3] Rest of the World (ROW) installed electrical power capacity, compiled from
www.data.UN.org (www.iea.com) and mecometer.com.

[4] ROW emissions estimated using an initial filling of 52 t/GW and a 12% loss during
manufacture and installation of new equipment and assuming 3% loss from banked $SF_6$ in
electrical equipment in 1980 and then decreasing linearly to 1.0% in 2018, reflecting the
change from older to newer equipment (Olivier and Bakker, 1999).


Table 2. The East Asian setup of the inversion system run 2007-2018 in 2-yr blocks: The
Meteorology, Transport Model (ATM), geographical domains over which the ATM is run,
number of particles released, inversion time-steps, prior information and observations used.

| Inversion System | Atmospheric Transport Model | Driving Meteorology | Computational Domain | Inversion Domain | Particles Released | Release Time Step | Prior | Obs |
|---|---|---|---|---|---|---|---|---|
| InTEM | NAME | Unified Model 12-40 km horizontal | 54.3°E to 192.0°E  5.3°S to 74.3°N | 88.1°E to 145.9°E  16.0°N to 57.6°N | 20,000 hr$^{-1}$ | 2 hr | Population 2 kt over domain 300% uncertainty per sub-region | GSN |

Note. GSN=Gosan station, Korea.

Table 3. The European setup of each inversion system run each year 2013-2018: the
Meteorology, Transport Model (ATM), geographical domains over which the ATM's are run,
number of particles released, inversion time-steps, prior information and observations used.

| Inversion System | Atmospheric Transport Model | Driving Meteorology | Computational Domain | Inversion Domain | Particles Released | Release Time Step | Prior | Obs |
|---|---|---|---|---|---|---|---|---|
| InTEM | NAME | Unified Model 1.5 km nested in 12-17 km horizontal | 98.1°W to 39.6°E  10.6°N to 79.2°N | 14.3°W to 30.8°E  36.4°N to 66.3°N | 20,000 hr$^{-1}$ | 2 hr | Population 2 kt over domain 200% uncertainty per sub-region | MHD JFJ CMN TAC RGL BSD HFD |
| EBRIS | FLEXPART 9.1_Empa | ECMWF-IFS 0.2° x 0.2° (-4°E - 16°E, 39°N - 51°N) nested in 1° x 1° | Global | 12.0°W to 26.4°E  36.0°N to 62.0°N | 16,667 hr$^{-1}$ | 3 hr | Population 2kt over domain 100 % uncertainty for whole inversion domain | MHD JFJ CMN TAC |
| FLITS | FLEXPART 9.0 | ECMWF Operational 1° lat x 1° lon horizontal | Global | 20.0°W to 50.0°E  0.0°N to 80.0°N | 13,333 hr$^{-1}$ | 3 hr | Population 2 kt over domain 200% uncertainty per sub-region | MHD JFJ CMN TAC |


Note. Observing stations: MHD (Mace Head, Ireland); JFJ (Jungfraujoch, Switzerland);
CMN (Monte Cimone, Italy); TAC (Tacolneston, UK); RGL (Ridge Hill, UK); BSD
(Bilsdale, UK); HFD (Heathfield, UK).

Table 4. Global $SF_6$ mole fraction output from the AGAGE 12-box model.

| YEAR | Global mole fraction (ppt) | 16%ile | 84%ile | YEAR | Global mole fraction (ppt) | 16%ile | 84%ile |
|---|---|---|---|---|---|---|---|
| 1978 | 0.66 | 0.64 | 0.68 | 1999 | 4.34 | 4.25 | 4.43 |
| 1979 | 0.76 | 0.73 | 0.78 | 2000 | 4.53 | 4.43 | 4.63 |
| 1980 | 0.86 | 0.83 | 0.88 | 2001 | 4.73 | 4.62 | 4.83 |
| 1981 | 0.98 | 0.95 | 1.00 | 2002 | 4.94 | 4.83 | 5.04 |
| 1982 | 1.10 | 1.08 | 1.13 | 2003 | 5.17 | 5.06 | 5.27 |
| 1983 | 1.22 | 1.19 | 1.25 | 2004 | 5.39 | 5.28 | 5.50 |
| 1984 | 1.34 | 1.31 | 1.37 | 2005 | 5.62 | 5.50 | 5.74 |
| 1985 | 1.49 | 1.45 | 1.52 | 2006 | 5.87 | 5.74 | 5.99 |
| 1986 | 1.65 | 1.61 | 1.68 | 2007 | 6.14 | 6.00 | 6.26 |
| 1987 | 1.81 | 1.77 | 1.85 | 2008 | 6.42 | 6.27 | 6.54 |
| 1988 | 1.98 | 1.93 | 2.02 | 2009 | 6.70 | 6.55 | 6.84 |
| 1989 | 2.15 | 2.11 | 2.20 | 2010 | 6.99 | 6.83 | 7.12 |
| 1990 | 2.35 | 2.30 | 2.40 | 2011 | 7.28 | 7.12 | 7.42 |
| 1991 | 2.56 | 2.51 | 2.62 | 2012 | 7.59 | 7.42 | 7.73 |
| 1992 | 2.77 | 2.71 | 2.83 | 2013 | 7.90 | 7.73 | 8.06 |
| 1993 | 2.98 | 2.92 | 3.05 | 2014 | 8.23 | 8.05 | 8.39 |
| 1994 | 3.21 | 3.13 | 3.27 | 2015 | 8.56 | 8.37 | 8.73 |
| 1995 | 3.45 | 3.37 | 3.52 | 2016 | 8.89 | 8.69 | 9.06 |
| 1996 | 3.69 | 3.60 | 3.76 | 2017 | 9.24 | 9.03 | 9.42 |
| 1997 | 3.92 | 3.83 | 4.00 | 2018 | 9.59 | 9.37 | 9.79 |
| 1998 | 4.14 | 4.04 | 4.22 | | | | |

Note: Mole fractions are reported at the mid-point of the year.
Table 5. Modelled global and SF$_6$ emissions from China. (Gg yr$^{-1}$) and UNFCCC reported
emissions.

| YEAR | *This work Global Emissions (Gg yr$^{-1}$) | UNFCCC Annex-1 | UNFCCC Revised Non-Annex-1 | UNFCCC Combined Annex-1 + Revised Non-Annex-1 | InTEM: China (Gosan site) | Krol et al., 2018 |
|---|---|---|---|---|---|---|
| 1978 | 2.51 (2.11-2.83) | | | | | |
| 1979 | 2.58 (2.29-2.84) | | | | | |
| 1980 | 2.74 (2.47-2.93) | | | | | |
| 1981 | 3.10 (2.83-3.40) | | | | | |
| 1982 | 3.13 (2.86-3.41) | | | | | |
| 1983 | 2.98 (2.65-3.15) | | | | | |
| 1984 | 3.38 (3.08-3.60) | | | | | |
| 1985 | 3.90 (3.61-4.18) | | | | | |
| 1986 | 4.20 (3.92)4.42) | | | | | |
| 1987 | 4.19 (3.89-4.43) | | | | | |
| 1988 | 4.34 (4.05-4.59) | | | | | 4.30 |
| 1989 | 4.70 (4.41-4.97) | | | | | 4.33 |
| 1990 | 5.16 (4.85-5.46) | 2.62 | 0.08 | 2.71 | | 4.77 |
| 1991 | 5.43 (5.12-5.75) | 2.66 | 0.03 | 2.69 | | 5.14 |
| 1992 | 5.40 (5.07-5.63) | 2.63 | 0.03 | 2.66 | | 5.59 |
| 1993 | 5.52 (5.20-5.81) | 2.63 | 0.16 | 2.79 | | 6.00 |
| 1994 | 5.88 (5.49-6.14) | 2.60 | 0.06 | 2.66 | | 6.36 |
| 1995 | 6.19 (5.86-6.54) | 2.65 | 0.16 | 2.81 | | 6.41 |
| 1996 | 6.13 (5.75-6.35) | 2.65 | 0.11 | 2.76 | | 6.06 |
| 1997 | 5.84 (5.55-6.13) | 2.40 | 0.13 | 2.53 | | 5.56 |
| 1998 | 5.57 (5.31-5.82) | 2.22 | 0.08 | 2.29 | | 5.35 |
| 1999 | 5.21 (4.94-5.49) | 1.93 | 0.17 | 2.09 | | 5.42 |
| 2000 | 5.04 (4.68-5.32) | 1.72 | 0.29 | 2.01 | | 5.55 |
| 2001 | 5.21 (4.89-5.43) | 1.58 | 0.20 | 1.78 | | 5.51 |
| 2002 | 5.61 (5.31-5.89) | 1.47 | 0.35 | 1.82 | | 5.63 |
| 2003 | 5.81 (5.53-6.03) | 1.40 | 0.40 | 1.80 | | 5.79 |
| 2004 | 5.83 (5.57-6.05) | 1.33 | 0.48 | 1.81 | | 5.86 |

| 2005 | 6.11 (5.87-6.35) | 1.25 | 0.99 | 2.23 | | 5.98 |
|------|------------------|------|------|------|--------------------|------|
| 2006 | 6.50 (6.21-6.72) | 1.18 | 0.94 | 2.12 | | 6.29 |
| 2007 | 6.98 (6.64-7.20) | 1.04 | 1.22 | 2.26 | 1.40 (1.01-1.79) | 6.79 |
| 2008 | 7.25 (6.89-7.45) | 0.94 | 1.22 | 2.16 | 1.48 (1.13-1.83) | 7.18 |
| 2009 | 7.20 (6.92-7.47) | 0.78 | 1.28 | 2.06 | 1.64 (1.31-1.96) | 7.26 |
| 2010 | 7.37 (7.05-7.65) | 0.79 | 1.77 | 2.56 | 1.77 (1.40-2.13) | 7.36 |
| 2011 | 7.65 (7.35-7.98) | 0.80 | 1.69 | 2.49 | 2.41 (2.01-2.82) | 7.56 |
| 2012 | 7.95 (7.59-8.20) | 0.95 | 1.76 | 2.71 | 2.57 (2.20-2.95) | 7.78 |
| 2013 | 8.20 (7.86-8.50) | 0.91 | 2.62 | 3.53 | 2.02 (1.65-2.40) | 7.96 |
| 2014 | 8.39 (8.05-8.65) | 0.72 | 3.22 | 3.94 | 2.09 (1.66-2.53) | 8.16 |
| 2015 | 8.45 (8.11-8.73) | 0.72 | 3.41 | 4.13 | 2.52 (2.03-3.02) | 8.36 |
| 2016 | 8.73 (8.37-8.99) | 0.75 | 4.18 | 4.93 | 2.81 (2.24-3.38) | 4.30 |
| 2017 | 8.92 (8.56-9.24) | NR | NR | NR | 2.71 (2.09-3.32) | 4.33 |
| 2018 | 9.04 (8.63-9.34) | NR | NR | NR | 3.22 (2.64-3.81) | 4.77 |

Note: Global emissions are mid-year. NR = Not reported. Revised Non-Annex-1 includes
interpolated values for missing years. Uncertainties shown in parenthesis as 16%ile and
84%ile. China $SF_6$ emissions estimated by InTEM were scaled to total China emissions by
population.














Table 6.  SF$_6$ emission estimates for Western Europe; UNFCCC inventory and InTEM,
EMPA and FLITS emissions (Gg yr$^{-1}$). (Uncertainties in parenthesis).

| | Inventory 1yr | InTEM (3sites, 2yr) | InTEM (7sites, 1yr) | EMPA (4sites, 1yr) | FLITS (4sites, 1yr) |
|---|---|---|---|---|---|
| 1990 | 0.48 (0.43-0.53) | | | | |
| 1991 | 0.5 (0.45-0.55) | | | | |
| 1992 | 0.54 (0.48-0.59) | | | | |
| 1993 | 0.57 (0.51-0.62) | | | | |
| 1994 | 0.61 (0.55-0.68) | | | | |
| 1995 | 0.66 (0.59-0.72) | | | | |
| 1996 | 0.65 (0.58-0.71) | | | | |
| 1997 | 0.58 (0.52-0.64) | | | | |
| 1998 | 0.55 (0.50-0.61) | | | | |
| 1999 | 0.45 (0.41-0.50) | | | | |
| 2000 | 0.45 (0.41-0.50) | | | | |
| 2001 | 0.42 (0.37-0.46) | | | | |
| 2002 | 0.37 (0.33-0.40) | | | | |
| 2003 | 0.34 (0.31-0.38) | | | | |
| 2004 | 0.35 (0.31-0.38) | | | | |
| 2005 | 0.33 (0.30-0.37) | | | | |
| 2006 | 0.31 (0.28-0.35) | | | | |
| 2007 | 0.29 (0.26-0.32) | 0.25 (0.16-0.34) | | | |
| 2008 | 0.28 (0.26-0.31) | 0.23 (0.16-0.31) | | | |
| 2009 | 0.27 (0.24-0.29) | 0.19 (0.13-0.25) | | | |
| 2010 | 0.27 (0.24-0.29) | 0.19 (0.13-0.25) | | | |
| 2011 | 0.26 (0.23-0.28) | 0.21 (0.16-0.27) | | | |
| 2012 | 0.26 (0.24-0.29) | 0.21 (0.16-0.27) | | | |
| 2013 | 0.26 (0.23-0.28) | 0.22 (0.16-0.28) | 0.26 (0.19-0.33) | 0.32 (0.29-0.35) | 0.37 (0.27-0.46) |
| 2014 | 0.25 (0.23-0.28) | 0.23 (0.16-0.29) | 0.28 (0.22-0.33) | 0.24 (0.21-0.27) | 0.36 (0.26-0.45) |
| 2015 | 0.26 (0.24-0.29) | 0.22 (0.16-0.29) | 0.29 (0.22-0.36) | 0.32 (0.30-0.34) | 0.37 (0.27-0.47) |
| 2016 | 0.27 (0.24-0.30) | 0.22 (0.15-0.30) | 0.27 (0.21-0.33) | 0.24 (0.21-0.27) | 0.34 (0.25-0.43) |
| 2017 | 0.28 (0.25-0.31) | 0.21 (0.14-0.28) | 0.30 (0.20-0.39) | 0.36 (0.32-0.40) | 0.45 (0.33-0.56) |
| 2018 | NR | 0.21 (0.15-0.27) | 0.23 (0.18-0.28) | 0.20 (0.17-0.23) | 0.30 (0.19-0.42) |

NR= Not reported,


## Author contributions

S.P., S.O'D., P.B.K., P.J.F., L.P.S., B.M., S.H., S.R., M.M., J.A., J.M., R.F.W., C.M.H.,
M.K.V., M.P., H.P., T.A., D.Y., and C.R contributed observational data. M.R., A.J.M., F.G.,
and R.G.P., carried out atmospheric model simulations and inverse analysis with support
from P.K.S., and R.H.J.W. The authors A.Mc., and K.M.S., made valuable data analysis
contributions to the work.


**Competing interests**

We confirm that there are no competing interests.

**Data availability**

The entire ALE/GAGE/AGAGE data base comprising every calibrated measurement including pollution events is archived on the official AGAGE website http://agage.mit.edu/data (note guidelines for use of AGAGE data), and on the ESS-DIVE website http://cdiac.ess-dive.lbl.gov/ndps/alegage.html. UK DECC data is available from the UK Natural Environment Research Council's (NERC) Centre for Environmental Data Analysis.

**Acknowledgements.**

We specifically acknowledge the cooperation and efforts of the station operators (G. Spain, MHD; R. Dickau, THD; P. Sealy, RPB; NOAA officer-in-charge, SMO) at the AGAGE stations and all other station managers and support staff at the different monitoring sites used in this study. We particularly thank NOAA and NILU for supplying some of the archived air samples shown, allowing us to fill important gaps. The operation of the AGAGE stations was supported by the National Aeronautical and Space Administration (NASA, USA) (grants NAG5-12669, NNX07AE89G, NNX11AF17G and NNX16AC98G to MIT; grants NAG5-4023, NNX07AE87G, NNX07AF09G, NNX11AF15G and NNX11AF16G to SIO); the Department for Business, Energy & Industrial Strategy (BEIS, UK formerly the Department of Energy and Climate Change (DECC)) (contracts GA01103, 1028/06/2015, 1537/06/2018 to the University of Bristol); the National Oceanic and Atmospheric Administration (NOAA, USA) (contract RA-133R-15-CN-0008 to the University of Bristol for Barbados, in addition to the operations of American Samoa station); and the Commonwealth Scientific and Industrial Research Organisation (CSIRO, Australia), Bureau of Meteorology (Australia), for their ongoing long-term support of the Cape Grim station and the Cape Grim science program. From 2017 HFD measurements were supported by the UK National Measurement System research funding to NPL. he measurements at Gosan, South Korea were supported by the Basic Science Research Program through the National Research Foundation of Korea (NRF) funded by the Ministry of Education (No. NRF-2019R1A2B5B02070239). Financial support for the Zeppelin measurements is acknowledged from the Norwegian Environment Agency. Financial support for the Jungfraujoch measurements is acknowledged from the Swiss national programmes HALCLIM and CLIMGAS-CH (Swiss Federal Office for the Environment, FOEN) as well as by ICOS-CH (Integrated Carbon Observation System Research Infrastructure). Support for the Jungfraujoch station was provided by International Foundation High Altitude Research Stations Jungfraujoch and Gornergrat (HFSJG). M. Rigby is supported by a NERC Advanced Fellowship NE/I021365/1. We also thank Dr R Derwent for valuable discussions on certain aspects of the paper.

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
