# Peer review of "The increasing atmospheric burden of the greenhouse gas sulfur hexafluoride (SF₆)"

_Atmospheric Chemistry and Physics, 2020_

## Referee Comment (RC1) · Anonymous Referee #1 · 24 Mar 2020

This manuscript bring together SF6 data since 1978 to present (2018) and discusses the global total and regional emissions based on inversion modelling tools. The research topic is relevant and executed nicely. I have enjoyed reading the manuscript which is prepared in a timely manner. I have a few comments for consideration by the authors, before the manuscript can be accepted for publication. I reallly want the authors to explain the anomalies seen the time series of global growth rates, global total emissions and emission from China.

Specific comments:

Should you add at the end of the first para in Abstract: "and demands in the Annex 1 countries"? I think it is becoming clear that the manufacturing is moving from the Annex 1 countries to the non-Annex 1 contries, and that is the main reason for the fast

rise in power grid development, say in China. Should consumer countries take some part of the responsibility?

lines 78-88: Jochen Harnisch also did some good work using SF6 from Hyderabad and NH high latitudes. May be cited here with a context SF6 loss are likely to quite fast in the high latitudes (Harnish et al., GRL, 1996). More recently Patra et al. (SOLA, 2018) have clearly shown that indeed the SF6 loss in the stratosphere and mesosphere is troublesome, by comparing with model simulations and CO2 profiles. How much will the stratospheric/mesopheric loss affect the global total budget calculatin of SF6 is relevant for this discussion. Hopefully this will be addressed in the future.

lines 126-127: What about Lovelock's work?

line 378: Are the table numbering in Sequence? please confirm.

Figure 1 & 2 and associated text: The change in growth rate during ~1995-2000 should be discussed, even if it is well known. Several aspects need attention: Quality of data continuity, inversion differences, EDGAR vs inversion differences.

How does the global total numbers compares with other studies, e.g., TransCom-AoA (Krol et al., 2018). Although they did not run an explicit inversion, their simulations are consistent with global mean SF6 growth rates, and should be comparable here. Such a comparison may help establishing a consistency between the 12-box model and 3-D global models

line 430: Is there a reference for this? Why has this to be from the SH but not from Southern areas of Pacific?

Figure 4 and associated discussion: y-axis units - Gg/yr?

The results of Zhou et al. (2018) is likely impossible given that the global emissions are in the range of ~8 Gg/yr.

How do you explain hump you get by InTEM T-D? The limitations of inverse modelling,

if any, should be addressed otherwise.

How is this (hump) consistent with your global total results in Fig. 2

line 580-583: Interesting twist to the stories of the renewables. Is there are good reason not to have a better Switches for the green tech power distribution?

Figure 8 and discussions: Apparently, you are trying to explain the faster increase in recent SF6 emission increase by the smaller electricity facilities. How many times more leaks the Wind and Solar power grids should have to explain the sharp rise in emissions since the 2000s? Noting that the electrical equipments remain functional for 30-40 years, the installations since 2000 must be working fine even today for Wind and Solar installations.

lines 635-638: Is this because there is some international standard in making the GIS - most are equally good? In principle it is hard to believe the new installation of worst quality than those in the Annex-1 countries only offset, not vastly exceed the emission reductions in Europe and USA.

---

## Referee Comment (RC2) · Anonymous Referee #2 · 8 Apr 2020

This paper describes the surface mole fraction and emission history over the last four decades of SF6 based primarily on AGAGE station measurements, inversion modeling and industry reports. This is a comprehensive analysis of the current knowledge of SF6 emissions and how they have shifted from Annex-1 to non-Annex-1 countries in the last decade. The manuscript is well written and the techniques are clearly defined. I recommend publication in ACP after consideration of the minor comments listed below.

Specific comments:

Lines 78-88: This paragraph is an odd fit here since it includes too much detail. If the point is to quote the Patra et al. (1997) lifetime estimates then I would just include that with the Ray et al. and Kovacs et al. estimates in the prior paragraph. The profile shape and correlations with other tracers in the stratosphere aren't really relevant here

since those details are well established and can be found in the cited references that
derived the atmospheric lifetimes.

Line 175: 'through' instead of 'though' and maybe spell out 'five' so a dash isn't necessary before 'core'.

Line 329: 'resolved'

Lines 397-403: Are the large differences between the bottom up EDGAR and UNFCCC
estimates easily explained? If so, it might be nice to include a brief statement on the
reason(s) here.

Line 434: The values shown in Fig. 4 and Table 5 are actually the scaled emissions
for all of China so that should be made clear here. As stated, it reads that the values
shown are only for Eastern mainland China.

Lines 450-456: It seems like you're referring to the same UNFCCC black symbols in
these sentences so it reads a little awkward.

Figure 4: I'm not sure if it's just my version but the axes on this figure are barely visible.

Line 503: It would be better to consistently refer to either FLITS or Urbino in the text
and Figures 6 and 7.

Lines 511-512: '. . .emissions to the global total in 2018 was 3.1% (2.4-3.9%, Table 6,
average of all inversions).'

Figure 8: The inset figure axes labels are so small they are difficult to read.

Line 659: add comma after '1978'

Line 672: remove comma before 'countries'

Table 5: Should include the population scaling factor here even if it is also in the text.

---

## Author Comment (AC1) · 30 Apr 2020

We thank Anonymous Referee # 1 for his diligence in reading our SF₆ paper and the constructive comments and suggestions. Our specific responses are listed below in bold text.

Should you add at the end of the first para in Abstract: "and demands in the Annex 1 countries"? I think it is becoming clear that the manufacturing is moving from the Annex 1 countries to the non-Annex 1 countries, and that is the main reason for the fast rise in power grid development, say in China. Should consumer countries take some part of the responsibility?

**RESPONSE**

**Added to line 54, before last sentence.**

**"Nevertheless, there are still demands for SF6 in Annex-1 countries due to economic growth, as well as continuing emissions from older equipment and additional emissions from newly installed SF6-insulated electrical equipment, although at low emissions rates".**

lines 78-88: Jochen Harnisch also did some good work using SF6 from Hyderabad and NH high latitudes. May be cited here with a context SF6 loss are likely to quite fast in the high latitudes (Harnish et al., GRL, 1996). More recently Patra et al. (SOLA, 2018) have clearly shown that indeed the SF6 loss in the stratosphere and mesosphere is troublesome, by comparing with model simulations and CO2 profiles. How much will the stratospheric/mesospheric loss affect the global total budget calculation of SF6 is relevant for this discussion. Hopefully this will be addressed in the future.

**RESPONSE**

**We thank the referee for pointing out the Harnisch et al., GRL, 1996 and Patra et al. (SOLA 2018) publications which have been included. We have edited and revised lines 78-88 to clarify this section and moved lines 76-77 as follows:-**

**"Vertical profiles of SF₆ mixing ratios, collected from balloon flights up to an altitude of about 37 km, indicated that there is very little loss of SF₆ due to photochemistry in the troposphere and lower stratosphere (Harnish et al., 1996; Patra et al., (1997). Using an improved atmospheric-chemical-transport model (Patra et al., 2018) reported significantly older 'age of air' (AoA) in the stratosphere and Krol et al., (2018), based on a comparison of six global transport models showed that upper stratospheric AOA varied from 4-7 years among the models.**

**It has been suggested that SF₆ may have a shorter atmospheric lifetime ranging from 1937 ± 432 years (Patra et al.,1997), 580-1400 years (Ray et al., 2017) and 1120-1475 years (Kovács et al., 2017). However, these shorter, but still very long, SF₆ lifetimes would not significantly affect SF₆ emissions estimated from atmospheric trends (Engel and Rigby, 2019) .Given the very long lifetime of SF6, compared to the period of our study, uncertainties in this term had a small influence on the outcome. For example, changing the lifetime from 3000 to 1000 years changed the derived emissions by around 1%, which is smaller than the derived uncertainties".**

lines 126-127: What about Lovelock's work?

**RESPONSE**

**We thank the referee for reminding us of the Lovelock reference. This has now been added to the paper.**

**"Lovelock, J.E. Atmospheric fluorine compounds as indicators of air movements. (Nature 230, 379, 1971) ".**

line 378: Are the table numbering in Sequence? please confirm.

**RESPONSE**

**Confirm Table numbering is in sequence.**

Figure 1 & 2 and associated text: The change in growth rate during ~1995-2000 should be discussed, even if it is well known. Several aspects need attention: Quality of data continuity, inversion differences, EDGAR vs inversion differences.

**RESPONSE**

**Please note that we discussed the change in growth rate during 1995-2000, as described on lines 393-396 and have also added the following additional text on line 136.**

**"However, after 1995 the annual average growth rate from 1996-2000 declined by 12.5% to 0.21 ppt $yr^{-1}$, coincident with a ~ 32% decrease in annual sales and prompt releases of $SF_6$ over this same time period (as noted in Table S2 of the Rand report)".**

**Inversion differences with EDGAR and Levin et al. (2010) are mentioned on line 387. In addition to this we have expanded the text :-**

**"Our estimates are in close agreement through 2008 with the independent top-down estimates of Levin et al. (2010). Our estimates show similar trends to EDGAR v4.2, although our global total is on average 8.9% higher. It should be noted that the EDGAR estimate includes some information from atmospheric observations (Rigby et al., 2010). On the other hand, it is likely that Annex-I countries are underreporting to the UNFCCC (Weiss and Prinn, 2011) and non-Annex-I countries are not required to report to UNFCCC which explains the much lower UNFCCC totals".**

**With regard to data quality and continuity we refer the referee to Section 2.1, which describes calibration and archiving of AGAGE data. More details are given in the key reference (Prinn et al., 2018).**

How does the global total numbers compares with other studies, e.g., TransCom-AoA (Krol et al., 2018). Although they did not run an explicit inversion, their simulations are consistent with global mean SF6 growth rates, and should be comparable here. Such a comparison may help establishing a consistency between the 12-box model and 3-D global models.

**RESPONSE**

**We have included the modelled results from Krol et al., 2018 (TransCom-AoA) in Table 5 and added the following text on line 390.**

**"There is close agreement, within the uncertainties, of our modelled global SF₆ emission estimates and those reported by Krol et al., (2018). The average annual difference was 0.2 Gg yr⁻¹ (range 0.01-0.49 Gg yr⁻¹) ".**

line 430: Is there a reference for this? Why has this to be from the SH but not from Southern areas of Pacific?

**RESPONSE**

**We have added a new reference, (S. Lui., 2011) and replaced the text on lines 430-431.**

**"GSN occasionally shows lower mixing ratios than MHD during the summer months when the monsoon transports oceanic background air from the Southern regions to the Gosan site on Jeju Island, South Korea".**

Figure 4 and associated discussion: y-axis units - Gg/yr?

**RESPONSE**

**Changed y-axis to Gg yr⁻¹**

The results of Zhou et al. (2018) is likely impossible given that the global emissions are in the range of ~8 Gg/yr.

**RESPONSE**

**Added this sentence on line 450.**

**"Notably from 2007-2012 the bottom-up estimates, after Zhou et al., (2018), with a FF of (52 t/GW) and high EF are in close agreement with the top-down estimates of Fang et al., (2014), They also agree with our results within uncertainties. However, after about 2014 the increase in these bottom-up estimates especially with the highest FF (66 t/GW), appear to represent an unrealistically large percentage of global emissions".**

How do you explain hump you get by InTEM T-D? The limitations of inverse modelling, if any, should be addressed otherwise.

**RESPONSE**

**It is important to not over-interpret inverse modelling results and especially the mean value, the error bars show that a smooth growth across all of the years is also possible, i.e. it is possible, within 1 sigma, for the hump not to exist at all, it is also possible that the true emission is outside 1 sigma uncertainty, say at the 2 sigma level. We have added the following sentence for clarity.**

**'The InTEM results show an increasing emission from China, the temporary rise in the mean value in 2011-2012 needs to be understood within the context of the uncertainty estimates, it is plausible, within 1 sigma, that there was no enhanced emissions during this period.'**

How is this (hump) consistent with your global total results in Fig. 2

**RESPONSE**

**See previous response**

line 580-583: Interesting twist to the stories of the renewables. Is there are good reason not to have a better Switches for the green tech power distribution?

**RESPONSE**

**We have added the following sentence for clarity.**

**"With the adoption of more technologically advanced GIS with lower emissions we might expect there to be a reduction in overall SF$_6$ emissions over time".**

Figure 8 and discussions: Apparently, you are trying to explain the faster increase in recent SF6 emission increase by the smaller electricity facilities. How many times more leaks the Wind and Solar power grids should have to explain the sharp rise in emissions since the 2000s? Noting that the electrical equipment remains functional for 30-40 years, the installations since 2000 must be working fine even today for Wind and Solar installations.

**RESPONSE**

**We have added the following sentences for clarity after the last sentence on page 17.**

**"There is clearly a balance between the very substantial increase in the global number of newly installed GIS equipment and the major advances in reducing the leakage of SF$_6$ from GIS equipment and the recovery and substitution of SF$_6$. At present the larger number of global GIS installations appear to be overpowering the success in reducing SF$_6$ emissions".**

lines 635-638: Is this because there is some international standard in making the GIS - most are equally good? In principle it is hard to believe the new installation of worst quality than those in the Annex-1 countries only offset, not vastly exceed the emission reductions in Europe and USA.

**RESPONSE**

**We have revised lines 635-638 as follows:-**

**"This suggests that when considered on ~10 year timescales, reductions in EF achieved in certain countries through new technologies or improved GIS management, have been offset by the very substantial growth in newly installed GIS with comparable EFs from other parts of the world, such that the effective EF has not changed substantially on a global scale".**

---

## Author Comment (AC2) · 30 Apr 2020

**We thank Anonymous Referee # 2 for his valuable constructive comments and suggestions. Our specific responses are listed below in bold text.**

Lines 78-88: This paragraph is an odd fit here since it includes too much detail. If the point is to quote the Patra et al. (1997) lifetime estimates then I would just include that with the Ray et al. and Kovacs et al. estimates in the prior paragraph. The profile shape and correlations with other tracers in the stratosphere aren't really relevant here since those details are well established and can be found in the cited references that derived the atmospheric lifetimes.

**RESPONSE**

**We have edited and re-ordered this section as follows: -.**

> **"Vertical profiles of $SF_6$ mixing ratios, collected from balloon flights up to an altitude of about 37 km, indicated that there is very little loss of $SF_6$ due to photochemistry in the troposphere and lower stratosphere (Harnish et al., 1996; Patra et al., (1997). Using an improved atmospheric-chemical-transport model (Patra et al., 2018) reported significantly older 'age of air' (AoA) in the stratosphere and Krol et al., (2018), based on a comparison of six global transport models showed that upper stratospheric AOA varied from 4-7 year among the models.**
>
> **It has been suggested that $SF_6$ may have a shorter atmospheric lifetime ranging from 1937 ± 432 years (Patra et al.,1997), 580-1400 years (Ray et al., 2017) and 1120-1475 years (Kovács et al., 2017). However, these shorter, but still very long, $SF_6$ lifetimes would not significantly affect $SF_6$ emissions estimated from atmospheric trends (Engel and Rigby, 2019) .Given the very long lifetime of SF6, compared to the period of our study, uncertainties in this term had a small influence on the outcome. For example, changing the lifetime from 3000 to 1000 years changed the derived emissions by around 1%, which is smaller than the derived uncertainties".**

Line 175: 'through' instead of 'though' and maybe spell out 'five' so a dash isn't necessary before 'core'.

**RESPONSE**

**Thank you, corrected.**

Line 329: 'resolved'

**RESPONSE**

**Corrected.**

Lines 397-403: Are the large differences between the bottom up EDGAR and UNFCCC estimates easily explained? If so, it might be nice to include a brief statement on the reason(s) here.

**RESPONSE**

**We have expanded the text on line 387 to address this point.**

**Our estimates are in close agreement through 2008 with the independent top-down estimates of Levin et al. (2010). Our estimates show similar trends to EDGAR v4.2, although our global total is on average 8.9% higher. It should be noted that the EDGAR**

**estimate includes some information from atmospheric observations (Rigby et al., 2010). On the other hand, it is likely that Annex-I countries are underreporting to the UNFCCC (Weiss and Prinn, 2011) and non-Annex-I countries are not required to report to UNFCCC which explains the much lower UNFCCC totals".**

Line 434: The values shown in Fig. 4 and Table 5 are actually the scaled emissions for all of China so that should be made clear here. As stated, it reads that the values shown are only for Eastern mainland China.

**RESPONSE**

**Agreed. We have noted in Figure 4 and Table 5 that emissions are scaled emissions for all of China.**

Lines 450-456: It seems like you're referring to the same UNFCCC black symbols in these sentences so it reads a little awkward. Figure 4: I'm not sure if it's just my version but the axes on this figure are barely visible.

**RESPONSE**

**We have revised lines 448-453 to remove the duplication as follows: -**

**"Our bottom-up estimated emissions, using the high EFs, are generally larger than the bottom-up estimated China emissions determined by Fang et al., 2013, while China estimates based on the lower EFs suggested by Zhou et al. (2018) are much lower than the other Chinese emission estimates".**

**Figure 4 axes have been darkened to improve visibility.**

Line 503: It would be better to consistently refer to either FLITS or Urbino in the text and Figures 6 and 7.

**RESPONSE**

**Agreed. FLITS has been used throughout.**

Lines 511-512: '. . .emissions to the global total in 2018 was 3.1% (2.4-3.9%, Table 6, average of all inversions).'

**RESPONSE**

**Thank you have added your revision.**

Figure 8: The inset figure axes labels are so small they are difficult to read.

**RESPONSE**

**Inset figure axes have been enlarged.**

Line 659: add comma after '1978'

**RESPONSE**

**Done**

Line 672: remove comma before 'countries'

**RESPONSE**

**Done**

Table 5: Should include the population scaling factor here even if it is also in the text.

**RESPONSE**

**Agreed and added.**